# Measuring the Intrinsic Dimension of Objective Landscapes

**Chunyuan Li** *
Duke University
cl319@duke.edu

**Heerad Farkhoor, Rosanne Liu, and Jason Yosinski**
Uber AI Labs
{heerad,rosanne,yosinski}@uber.com

## Abstract

Many recently trained neural networks employ large numbers of parameters to achieve good performance. One may intuitively use the number of parameters required as a rough gauge of the difficulty of a problem. But how accurate are such notions? How many parameters are really needed? In this paper we attempt to answer this question by training networks not in their native parameter space, but instead in a smaller, randomly oriented subspace. We slowly increase the dimension of this subspace, note at which dimension solutions first appear, and define this to be the *intrinsic dimension* of the objective landscape. The approach is simple to implement, computationally tractable, and produces several suggestive conclusions. Many problems have smaller intrinsic dimensions than one might suspect, and the intrinsic dimension for a given dataset varies little across a family of models with vastly different sizes. This latter result has the profound implication that once a parameter space is large enough to solve a problem, extra parameters serve directly to increase the dimensionality of the solution manifold. Intrinsic dimension allows some quantitative comparison of problem difficulty across supervised, reinforcement, and other types of learning where we conclude, for example, that solving the inverted pendulum problem is 100 times easier than classifying digits from MNIST, and playing Atari Pong from pixels is about as hard as classifying CIFAR-10. In addition to providing new cartography of the objective landscapes wandered by parameterized models, the method is a simple technique for constructively obtaining an upper bound on the minimum description length of a solution. A byproduct of this construction is a simple approach for compressing networks, in some cases by more than 100 times.

## 1 Introduction

Training a neural network to model a given dataset entails several steps. First, the network designer chooses a loss function and a network architecture for a given dataset. The architecture is then initialized by populating its weights with random values drawn from some distribution. Finally, the network is trained by adjusting its weights to produce a loss as low as possible. We can think of the training procedure as traversing some path along an *objective landscape*. Note that as soon as a dataset and network architecture are specified, the landscape in its entirety is completely determined. It is instantiated and frozen; all subsequent parameter initialization, forward and backward propagation, and gradient steps taken by an optimizer are just details of how the frozen space is explored.

Consider a network parameterized by $D$ weights. We can picture its associated objective landscape as a set of "hills and valleys" in $D$ dimensions, where each point in $\mathbb{R}^D$ corresponds to a value of the loss, *i.e.,* the elevation of the landscape. If $D = 2$, the map from two coordinates to one scalar loss can be easily imagined and intuitively understood by those living in a three-dimensional world with similar hills. However, in higher dimensions, our intuitions may not be so faithful, and generally we must be careful, as extrapolating low-dimensional intuitions to higher dimensions can lead to unreliable conclusions. The difficulty of understanding high-dimensional landscapes notwithstanding, it is the lot of neural network researchers to spend their efforts leading (or following?) networks

---

*Work performed as an intern at Uber AI Labs.

over these multi-dimensional surfaces. Therefore, any interpreted geography of these landscapes is valuable.

Several papers have shed valuable light on this landscape, particularly by pointing out flaws in common extrapolation from low-dimensional reasoning. Dauphin et al. (2014) showed that, in contrast to conventional thinking about getting stuck in local optima (as one might be stuck in a valley in our familiar $D = 2$), local critical points in high dimension are almost never valleys but are instead saddlepoints: structures which are "valleys" along a multitude of dimensions with "exits" in a multitude of other dimensions. The striking conclusion is that one has less to fear becoming hemmed in on all sides by higher loss but more to fear being waylaid nearly indefinitely by nearly flat regions. Goodfellow et al. (2015) showed another property: that paths directly from the initial point to the final point of optimization are often monotonically decreasing. Though dimension is high, the space is in some sense simpler than we thought: rather than winding around hills and through long twisting corridors, the walk could just as well have taken a straight line without encountering any obstacles, if only the direction of the line could have been determined at the outset.

In this paper we seek further understanding of the structure of the objective landscape by restricting training to random slices through it, allowing optimization to proceed in randomly generated subspaces of the full parameter space. Whereas standard neural network training involves computing a gradient and taking a step in the full parameter space ($\mathbb{R}^D$ above), we instead choose a random $d$-dimensional subspace of $\mathbb{R}^D$, where generally $d < D$, and optimize directly in this subspace. By performing experiments with gradually larger values of $d$, we can find the subspace dimension at which solutions first appear, which we call the measured *intrinsic dimension* of a particular problem. Examining intrinsic dimensions across a variety of problems leads to a few new intuitions about the optimization problems that arise from neural network models.

We begin in Sec. 2 by defining more precisely the notion of intrinsic dimension as a measure of the difficulty of objective landscapes. In Sec. 3 we measure intrinsic dimension over a variety of network types and datasets, including MNIST, CIFAR-10, ImageNet, and several RL tasks. Based on these measurements, we draw a few insights on network behavior, and we conclude in Sec. 4.

## 2 DEFINING AND ESTIMATING INTRINSIC DIMENSION

We introduce the intrinsic dimension of an objective landscape with an illustrative toy problem. Let $\theta^{(D)} \in \mathbb{R}^D$ be a parameter vector in a parameter space of dimension $D$, let $\theta_0^{(D)}$ be a randomly chosen initial parameter vector, and let $\theta_*^{(D)}$ be the final parameter vector arrived at via optimization.

Consider a toy optimization problem where $D = 1000$ and where $\theta^{(D)}$ optimized to minimize a squared error cost function that requires the first 100 elements to sum to 1, the second 100 elements to sum to 2, and so on until the vector has been divided into 10 groups with their requisite 10 sums. We may start from a $\theta_0^{(D)}$ that is drawn from a Gaussian distribution and optimize in $\mathbb{R}^D$ to find a $\theta_*^{(D)}$ that solves the problem with cost arbitrarily close to zero.

Solutions to this problem are highly redundant. With a little algebra, one can find that the manifold of solutions is a 990 dimensional hyperplane: from any point that has zero cost, there are 990 orthogonal directions one can move and remain at zero cost. Denoting as $s$ the dimensionality of the solution set, we define the intrinsic dimensionality $d_{\text{int}}$ of a solution as the codimension of the solution set inside of $\mathbb{R}^D$:

$$D = d_{\text{int}} + s \tag{1}$$

Here the intrinsic dimension $d_{\text{int}}$ is 10 (1000 = 10 + 990), with 10 corresponding intuitively to the number of constraints placed on the parameter vector. Though the space is large ($D = 1000$), the number of things one needs to get right is small ($d_{\text{int}} = 10$).

### 2.1 MEASURING INTRINSIC DIMENSION VIA RANDOM SUBSPACE TRAINING

The above example had a simple enough form that we obtained $d_{\text{int}} = 10$ by calculation. But in general we desire a method to measure or approximate $d_{\text{int}}$ for more complicated problems, including problems with data-dependent objective functions, e.g. neural network training. Random subspace optimization provides such a method.

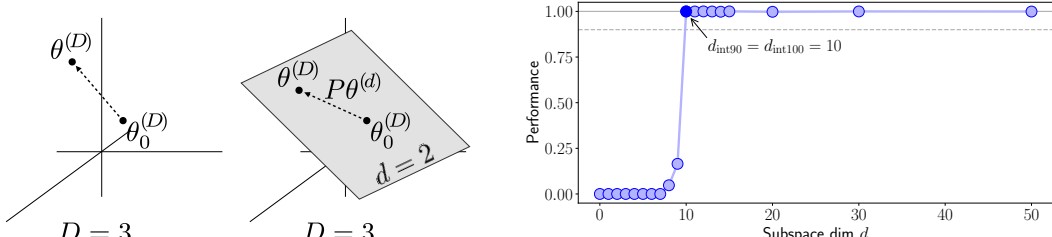

Figure 1: **(left)** Illustration of parameter vectors for direct optimization in the $D = 3$ case. **(middle)** Illustration of parameter vectors and a possible random subspace for the $D = 3, d = 2$ case. **(right)** Plot of performance *vs.* subspace dimension for the toy example of toy example of Sec. 2. The problem becomes both 90% solvable and 100% solvable at random subspace dimension 10, so $d_{\text{int}90}$ and $d_{\text{int}100}$ are 10.

Standard optimization, which we will refer to hereafter as the *direct* method of training, entails evaluating the gradient of a loss with respect to $\theta^{(D)}$ and taking steps directly in the space of $\theta^{(D)}$. To train in a random subspace, we instead define $\theta^{(D)}$ in the following way:

$$\theta^{(D)} = \theta_0^{(D)} + P\theta^{(d)} \qquad (2)$$

where $P$ is a randomly generated $D \times d$ projection matrix[1] and $\theta^{(d)}$ is a parameter vector in a generally smaller space $\mathbb{R}^d$. $\theta_0^{(D)}$ and $P$ are randomly generated and frozen (not trained), so the system has only $d$ degrees of freedom. We initialize $\theta^{(d)}$ to a vector of all zeros, so initially $\theta^{(D)} = \theta_0^{(D)}$. This convention serves an important purpose for neural network training: it allows the network to benefit from beginning in a region of parameter space designed by any number of good initialization schemes (Glorot & Bengio, 2010; He et al., 2015) to be well-conditioned, such that gradient descent via commonly used optimizers will tend to work well.[2]

Training proceeds by computing gradients with respect to $\theta^{(d)}$ and taking steps in that space. Columns of $P$ are normalized to unit length, so steps of unit length in $\theta^{(d)}$ chart out unit length motions of $\theta^{(D)}$. Columns of $P$ may also be orthogonalized if desired, but in our experiments we relied simply on the approximate orthogonality of high dimensional random vectors. By this construction $P$ forms an approximately orthonormal basis for a randomly oriented $d$ dimensional subspace of $\mathbb{R}^D$, with the origin of the new coordinate system at $\theta_0^{(D)}$. Fig. 1 (left and middle) shows an illustration of the related vectors.

Consider a few properties of this training approach. If $d = D$ and $P$ is a large identity matrix, we recover exactly the direct optimization problem. If $d = D$ but $P$ is instead a random orthonormal basis for all of $\mathbb{R}^D$ (just a random rotation matrix), we recover a rotated version of the direct problem. Note that for some "rotation-invariant" optimizers, such as SGD and SGD with momentum, rotating the basis will not change the steps taken nor the solution found, but for optimizers with axis-aligned assumptions, such as RMSProp (Tieleman & Hinton, 2012) and Adam (Kingma & Ba, 2014), the path taken through $\theta^{(D)}$ space by an optimizer will depend on the rotation chosen. Finally, in the general case where $d < D$ and solutions exist in $D$, solutions will *almost surely* (with probability 1) not be found if $d$ is less than the codimension of the solution. On the other hand, when $d \geq D - s$, if the solution set is a hyperplane, the solution will almost surely intersect the subspace, but for solution sets of arbitrary topology, intersection is not guaranteed. Nonetheless, by iteratively increasing $d$, re-running optimization, and checking for solutions, we obtain one estimate of $d_{\text{int}}$. We try this sweep of $d$ for our toy problem laid out in the beginning of this section, measuring (by convention

---

[1]This projection matrix can take a variety of forms each with different computational considerations. In later sections we consider dense, sparse, and implicit $P$ matrices.

[2]A second, more subtle reason to start away from the origin and with a randomly oriented subspace is that this puts the subspace used for training and the solution set in *general position* with respect to each other. Intuitively, this avoids pathological cases where both the solution set and the random subspace contain structure oriented around the origin or along axes, which could bias toward non-intersection or toward intersection, depending on the problem.

as described in the next section) the positive performance (higher is better) instead of loss.[3] As expected, the solutions are first found at $d = 10$ (see Fig. 1, right), confirming our intuition that for this problem, $d_{\text{int}} = 10$.

## 2.2 DETAILS AND CONVENTIONS

In the rest of this paper, we measure intrinsic dimensions for particular neural network problems and draw conclusions about the associated objective landscapes and solution sets. Because modeling real data is more complex than the above toy example, and losses are generally never exactly zero, we first choose a heuristic for classifying points on the objective landscape as solutions *vs.* non-solutions. The heuristic we choose is to threshold network performance at some level relative to a baseline model, where generally we take as baseline the best directly trained model. In supervised classification settings, validation accuracy is used as the measure of performance, and in reinforcement learning scenarios, the total reward (shifted up or down such that the minimum reward is 0) is used. Accuracy and reward are preferred to loss to ensure results are grounded to real-world performance and to allow comparison across models with differing scales of loss and different amounts of regularization included in the loss.

We define $d_{\text{int}100}$ as the intrinsic dimension of the "100%" solution: solutions whose performance is statistically indistinguishable from baseline solutions. However, when attempting to measure $d_{\text{int}100}$, we observed it to vary widely, for a few confounding reasons: $d_{\text{int}100}$ can be very high — nearly as high as $D$ — when the task requires matching a very well-tuned baseline model, but can drop significantly when the regularization effect of restricting parameters to a subspace boosts performance by tiny amounts. While these are interesting effects, we primarily set out to measure the basic difficulty of problems and the degrees of freedom needed to solve (or approximately solve) them rather than these subtler effects.

Thus, we found it more practical and useful to define and measure $d_{\text{int}90}$ as the intrinsic dimension of the "90%" solution: solutions with performance at least 90% of the baseline.

We chose 90% after looking at a number of dimension *vs.* performance plots (e.g. Fig. 2) as a reasonable trade off between wanting to guarantee solutions are as good as possible, but also wanting measured $d_{\text{int}}$ values to be robust to small noise in measured performance. If too high a threshold is used, then the dimension at which performance crosses the threshold changes a lot for only tiny changes in accuracy, and we always observe tiny changes in accuracy due to training noise.

If a somewhat different (higher or lower) threshold were chosen, we expect most of conclusions in the rest of the paper to remain qualitatively unchanged. In the future, researchers may find it useful to measure $d_{\text{int}}$ using higher or lower thresholds.

## 3 RESULTS AND DISCUSSION

### 3.1 MNIST

We begin by analyzing a fully connected (FC) classifier trained on MNIST. We choose a network with layer sizes 784–200–200–10, i.e. a network with two hidden layers of width 200; this results in a total number of parameters $D = 199,210$. A series of experiments with gradually increasing subspace dimension $d$ produce monotonically increasing performances, as shown in Fig. 2 (left). By checking the subspace dimension at which performance crosses the 90% mark, we measure this network's intrinsic dimension $d_{\text{int}90}$ at about 750.

**Some networks are very compressible.** A salient initial conclusion is that 750 is quite low. At that subspace dimension, only 750 degrees of freedom (0.4%) are being used and 198,460 (99.6%) unused to obtain 90% of the performance of the direct baseline model. A compelling corollary of this result is a simple, new way of creating and training compressed networks, particularly networks for applications in which the absolute best performance is not critical. To store this network, one need only store a tuple of three items: (*i*) the random seed to generate the frozen $\theta_0^{(D)}$, (*ii*) the

---

[3]For this toy problem we define $\text{performance} = \exp(-\text{loss})$, bounding performance between 0 and 1, with 1 being a perfect solution.

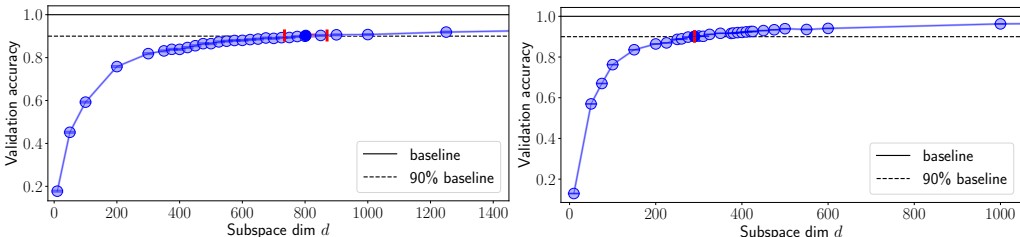

Figure 2: Performance (validation accuracy) *vs.* subspace dimension $d$ for two networks trained on MNIST: **(left)** a 784–200–200–10 fully-connected (FC) network ($D = 199{,}210$) and **(right)** a convolutional network, LeNet ($D = 44{,}426$). The solid line shows performance of a well-trained direct (FC or conv) model, and the dashed line shows the 90% threshold we use to define $d_{\text{int}90}$. The standard derivation of validation accuracy and measured $d_{\text{int}90}$ are visualized as the blue vertical and red horizontal error bars. We oversample the region around the threshold to estimate the dimension of crossing more exactly. We use one-run measurements for $d_{\text{int}90}$ of 750 and 290, respectively.

random seed to generate $P$ and (*iii*) the 750 floating point numbers in $\theta_*^{(d)}$. It leads to compression (assuming 32-bit floats) by a factor of $260\times$ from 793kB to only 3.2kB, or 0.4% of the full parameter size. Such compression could be very useful for scenarios where storage or bandwidth are limited, e.g. including neural networks in downloaded mobile apps or on web pages.

This compression approach differs from other neural network compression methods in the following aspects. (*i*) While it has previously been appreciated that large networks waste parameters (Dauphin & Bengio, 2013) and weights contain redundancy (Denil et al., 2013) that can be exploited for post-hoc compression (Wen et al., 2016), this paper's method constitutes a much simpler approach to compression, where training happens once, end-to-end, and where any parameterized model is an allowable base model. (*ii*) Unlike layerwise compression models (Denil et al., 2013; Wen et al., 2016), we operate in the entire parameter space, which could work better or worse, depending on the network. (*iii*) Compared to methods like that of Louizos et al. (2017), who take a Bayesian perspective and consider redundancy on the level of groups of parameters (input weights to a single neuron) by using group-sparsity-inducing hierarchical priors on the weights, our approach is simpler but not likely to lead to compression as high as the levels they attain. (*iv*) Our approach only reduces the number of degrees of freedom, not the number of bits required to store each degree of freedom, e.g. as could be accomplished by quantizing weights (Han et al., 2016). Both approaches could be combined. (*v*) There is a beautiful array of papers on compressing networks such that they also achieve computational savings during the forward pass (Wen et al., 2016; Han et al., 2016; Yang et al., 2015); subspace training does not speed up execution time during inference. (*vi*) Finally, note the relationships between weight pruning, weight tying, and subspace training: weight pruning is equivalent to finding, post-hoc, a subspace that is orthogonal to certain axes of the full parameter space and that intersects those axes at the origin. Weight tying, e.g. by random hashing of weights into buckets (Chen et al., 2015), is equivalent to subspace training where the subspace is restricted to lie along the equidistant "diagonals" between any axes that are tied together.

**Robustness of intrinsic dimension.** Next, we investigate how intrinsic dimension varies across FC networks with a varying number of layers and varying layer width.[4] We perform a grid sweep of networks with number of hidden layers $L$ chosen from $\{1, 2, 3, 4, 5\}$ and width $W$ chosen from $\{50, 100, 200, 400\}$. Fig. S6 in the Supplementary Information shows performance *vs.* subspace dimension plots in the style of Fig. 2 for all 20 networks, and Fig. 3 shows each network's $d_{\text{int}90}$ plotted against its native dimension $D$. As one can see, $D$ changes by a factor of $24.1$ between the smallest and largest networks, but $d_{\text{int}90}$ changes over this range by a factor of only $1.33$, with much of this possibly due to noise.

Thus it turns out that the intrinsic dimension changes little even as models grown in width or depth! The striking conclusion is that every extra parameter added to the network — every extra dimension added to $D$ — just ends up adding one dimension to the redundancy of the solution, $s$.

---

[4]Note that here we used a global baseline of 100% accuracy to compare simply and fairly across all models. See Sec. S5 for similar results obtained using instead 20 separate baselines for each of the 20 models.

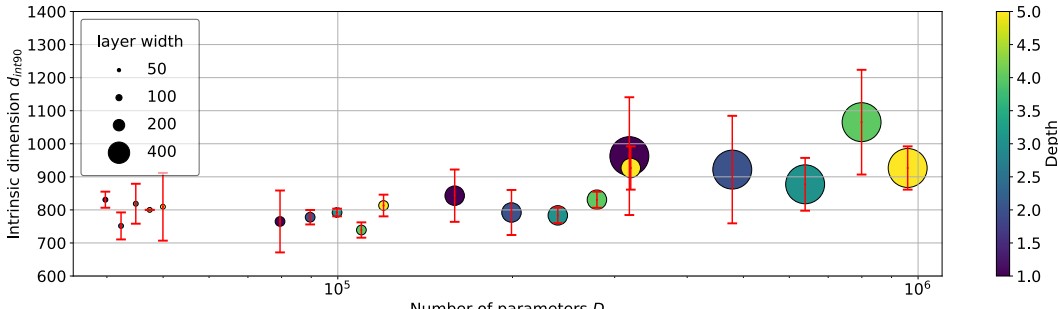

Figure 3: Measured intrinsic dimension $d_{\text{int}90}$ vs number of parameters $D$ for 20 FC models of varying width (from 50 to 400) and depth (number of hidden layers from 1 to 5) trained on MNIST. The red interval is the standard derivation of the measurement of $d_{\text{int}90}$. Though the number of native parameters $D$ varies by a factor of 24.1, $d_{\text{int}90}$ varies by only 1.33, with much of that factor possibly due to noise, showing that $d_{\text{int}90}$ is a fairly robust measure across a model family and that each extra parameter ends up adding an extra dimension directly to the redundancy of the solution. Standard deviation was estimated via bootstrap; see Sec. S5.1.

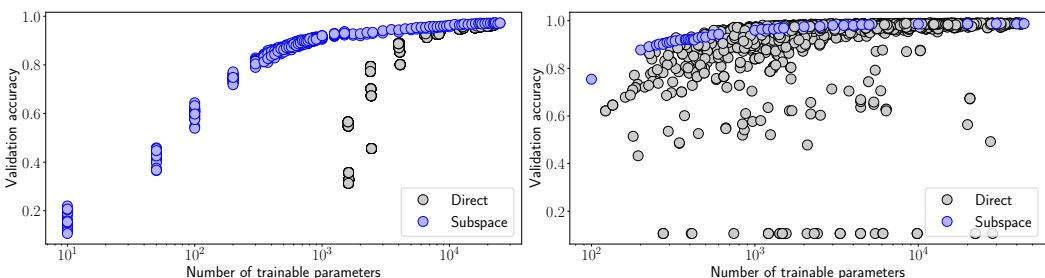

Figure 4: Performance *vs.* number of trainable parameters for **(left)** FC networks and **(right)** convolutional networks trained on MNIST. Randomly generated direct networks are shown (gray circles) alongside all random subspace training results (blue circles) from the sweep shown in Fig. S6. FC networks show a persistent gap in dimension, suggesting general parameter inefficiency of FC models. The parameter efficiency of convolutional networks varies, as the gray points can be significantly to the right of or close to the blue manifold.

Often the most accurate directly trained models for a problem have far more parameters than needed (Zhang et al., 2017); this may be because they are just easier to train, and our observation suggests a reason why: with larger models, solutions have greater redundancy and in a sense "cover" more of the space.[5] To our knowledge, this is the first time this phenomenon has been directly measured. We should also be careful not to claim that all FC nets on MNIST will have an intrinsic dimension of around 750; instead, we should just consider that we have found for this architecture/dataset combination a wide plateau of hyperparamter space over which intrinsic dimension is approximately constant.

**Are random subspaces really more parameter-efficient for FC nets?** One might wonder to what extent claiming 750 parameters is meaningful given that performance achieved (90%) is far worse than a state of the art network trained on MNIST. With such a low bar for performance, could a directly trained network with a comparable number of trainable parameters be found that achieves the same performance? We generated 1000 small networks (depth randomly chosen from {1, 2, 3, 4, 5}, layer width randomly from {2, 3, 5, 8, 10, 15, 20, 25}, seed set randomly) in an attempt to

---

[5]To be precise, we may not conclude "greater coverage" in terms of the volume of the solution set — volumes are not comparable across spaces of different dimension, and our measurements have only estimated the dimension of the solution set, not its volume. A conclusion we *may* make is that as extra parameters are added, the ratio of solution dimension to total dimension, $s/D$, increases, approaching 1. Further research could address other notions of coverage.

find high-performing, small FC networks, but as Fig. 4 (left) shows, a gap still exists between the subspace dimension and the smallest direct FC network giving the same performance at most levels of performance.

**Measuring $d_{\text{int90}}$ on a convolutional network.** Next we measure $d_{\text{int90}}$ of a convolutional network, LeNet (D=44,426). Fig. 2 (right) shows validation accuracy *vs.* subspace dimension $d$, and we find $d_{\text{int90}} = 290$, or a compression rate of about $150\times$ for this network. As with the FC case above, we also do a sweep of random networks,

but notice that the performance gap of convnets between direct and subspace training methods becomes closer for fixed budgets, *i.e.,* the number of trainable parameters. Further, the performance of direct training varies significantly, depending on the extrinsic design of convet architectures. We interpret these results in terms of the Minimum Description Length below.

**Relationship between Intrinsic Dimension and Minimum Description Length (MDL).** As discussed earlier, the random subspace training method leads naturally to a compressed representation of a network, where only $d$ floating point numbers need to be stored. We can consider this $d$ as an upper bound on the MDL of the problem solution.[6] We cannot yet conclude the extent to which this bound is loose or tight, and tightness may vary by problem. However, to the extent that it is tighter than previous bounds (*e.g.,* just the number of parameters $D$) and to the extent that it is correlated with the actual MDL, we can use this interpretation to judge which solutions are more well-suited to the problem in a principled way. As developed by Rissanen (1978) and further by Hinton & Van Camp (1993), holding accuracy constant, the best model is the one with the shortest MDL.

Thus, there is some rigor behind our intuitive assumption that LeNet is a *better* model than an FC network for MNIST image classification, because its intrinsic dimension is lower ($d_{\text{int90}}$ of 290 *vs.* 750). In this particular case we are lead to a predictable conclusion, but as models become larger, more complex, and more heterogeneous, conclusions of this type will often not be obvious. Having a simple method of approximating MDL may prove extremely useful for guiding model exploration, for example, for the countless datasets less well-studied than MNIST and for models consisting of separate sub-models that may be individually designed and evaluated (Ren et al., 2015; Kaiser et al., 2017). In this latter case, considering the MDL for a sub-model could provide a more detailed view of that sub-model's properties than would be available by just analyzing the system's overall validation performance.

Finally, note that although our approach is related to a rich body of work on estimating the "intrinsic dimension of a dataset" (Camastra & Vinciarelli, 2002; Kégl, 2003; Fukunaga & Olsen, 1971; Levina & Bickel, 2005; Tenenbaum et al., 2000), it differs in a few respects. Here we do not measure the number of degrees of freedom necessary to represent a dataset (which requires representation of a global $p(X)$ and per-example properties and thus grows with the size of the dataset), but those required to represent a model for part of the dataset (here $p(y|X)$, which intuitively might saturate at some complexity even as a dataset grows very large). That said, in the following section we do show measurements for a corner case where the model must memorize per-example properties.

**Are convnets *always* better on MNIST? Measuring $d_{\text{int90}}$ on shuffled data.** Zhang et al. (2017) provocatively showed that large networks normally thought to generalize well can nearly as easily be trained to memorize entire training sets with randomly assigned labels or with input pixels provided in random order. Consider two identically sized networks: one trained on a real, non-shuffled dataset and another trained with shuffled pixels or labels. As noted by Zhang et al. (2017), externally the networks are very similar, and the training loss may even be identical at the final epoch. However, the intrinsic dimension of each may be measured to expose the differences in problem difficulty. When training on a dataset with *shuffled pixels* — pixels for each example in the dataset subject to a random permutation, chosen once for the entire dataset — the intrinsic dimension of an FC network remains the same at 750, because FC networks are invariant to input permutation. But the intrinsic dimension of a convnet increases from 290 to 1400, even higher than an FC network. Thus while convnets are better suited to classifying digits given images with local structure, when this structure is removed, violating convolutional assumptions, our measure can clearly reveal that many

---

[6]We consider MDL in terms of number of degrees of freedom instead of bits. For degrees of freedom stored with constant fidelity (e.g. `float32`), these quantities are related by a constant factor (e.g. 32).

Table 1: Measured $d_{\text{int}90}$ on various supervised and reinforcement learning problems.

| Dataset | MNIST | | MNIST (Shuf Pixels) | | MNIST (Shuf Labels) |
|---|---|---|---|---|---|
| Network Type | FC | LeNet | FC | LeNet | FC |
| Parameter Dim. $D$ | 199,210 | 44,426 | 199,210 | 44,426 | 959,610 |
| Intrinsic Dim. $d_{\text{int}90}$ | 750 | 290 | 750 | 1,400 | 190,000 |

| | CIFAR-10 | | ImageNet | Inverted Pendulum | Humanoid | Atari Pong |
|---|---|---|---|---|---|---|
| ... | FC | LeNet | SqueezeNet | FC | FC | ConvNet |
| ... | 656,810 | 62,006 | 1,248,424 | 562 | 166,673 | 1,005,974 |
| ... | 9,000 | 2,900 | > 500k | 4 | 700 | 6,000 |

more degrees of freedom are now required to model the underlying distribution. When training on MNIST with *shuffled labels* — the label for each example is randomly chosen — we redefine our measure of $d_{\text{int}90}$ relative to training accuracy (validation accuracy is always at chance). We find that memorizing random labels on the 50,000 example MNIST training set requires a very high dimension, $d_{\text{int}90} = 190,000$, or 3.8 floats per memorized label. Sec. S5.2 gives a few further results, in particular that the more labels are memorized, the more efficient memorization is in terms of floats per label. Thus, while the network obviously does not generalize to an unseen validation set, it would seem "generalization" *within* a training set may be occurring as the network builds a shared infrastructure that makes it possible to more efficiently memorize labels.

## 3.2 CIFAR-10 AND IMAGENET

We scale to larger supervised classification problems by considering CIFAR-10 (Krizhevsky & Hinton, 2009) and ImageNet (Russakovsky et al., 2015). When scaling beyond MNIST-sized networks with $D$ on the order of 200k and $d$ on the order of 1k, we find it necessary to use more efficient methods of generating and projecting from random subspaces. This is particularly true in the case of ImageNet, where the direct network can easily require millions of parameters. In Sec. S7, we describe and characterize scaling properties of three methods of projection: dense matrix projection, sparse matrix projection (Li et al., 2006), and the remarkable Fastfood transform (Le et al., 2013). We generally use the sparse projection method to train networks on CIFAR-10 and the Fastfood transform for ImageNet.

Measured $d_{\text{int}90}$ values for CIFAR-10 and are ImageNet given in Table 1, next to all previous MNIST results and RL results to come. For CIFAR-10 we find qualitatively similar results to MNIST, but with generally higher dimension (9k *vs.* 750 for FC and 2.9k *vs.* 290 for LeNet). It is also interesting to observe the difference of $d_{\text{int}90}$ across network architectures. For example, to achieve a global >50% validation accuracy on CIFAR-10, FC, LeNet and ResNet approximately requires $d_{\text{int}90} = 9k$, 2.9k and 1k, respectively, showing that ResNets are more efficient. Full results and experiment details are given in Sec. S8 and Sec. S9. Due to limited time and memory issues, training on ImageNet has not yet given a reliable estimate for $d_{\text{int}90}$ except that it is over 500k.

## 3.3 REINFORCEMENT LEARNING ENVIRONMENTS

Measuring intrinsic dimension allows us to perform some comparison across the divide between supervised learning and reinforcement learning. In this section we measure the intrinsic dimension of three control tasks of varying difficulties using both value-based and policy-based algorithms. The value-based algorithm we evaluate is the Deep Q-Network (DQN) (Mnih et al., 2013), and the policy-based algorithm is Evolutionary Strategies (ES) (Salimans et al., 2017). Training details are given in Sec. S6.2. For all tasks, performance is defined as the maximum-attained (over training iterations) mean evaluation reward (averaged over 30 evaluations for a given parameter setting). In Fig. 5, we show results of ES on three tasks: `InvertedPendulum−v1`, `Humanoid−v1` in Mu-JoCo (Todorov et al., 2012), and `Pong−v0` in Atari. Dots in each plot correspond to the (noisy) median of observed performance values across many runs for each given $d$, and the vertical uncertainty bar shows the maximum and minimum observed performance values. The dotted horizontal line corresponds to the usual 90% baseline derived from the best directly-trained network (the solid

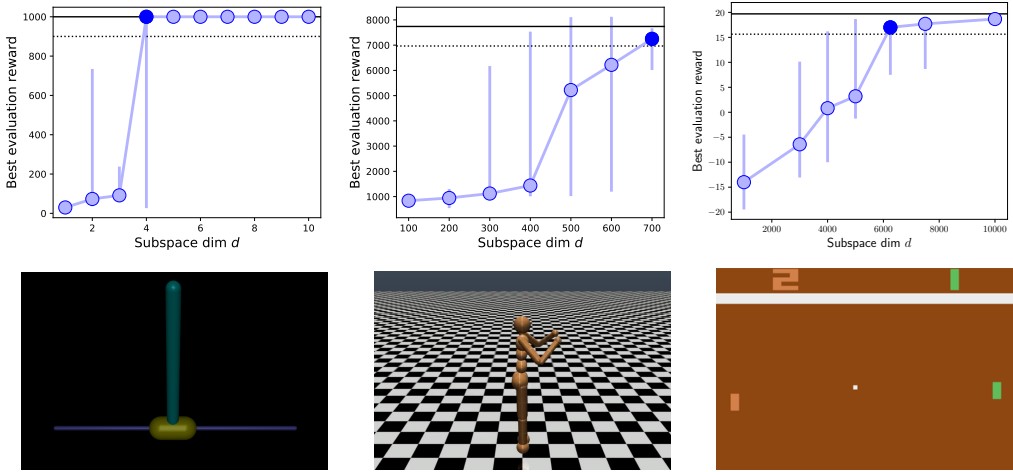

Figure 5: Results using the policy-based ES algorithm to train agents on **(left column)** `InvertedPendulum−v1`, **(middle column)** `Humanoid−v1`, and **(right column)** `Pong−v0`. The intrinsic dimensions found are $4$, $700$, and 6k. This places the walking humanoid task on a similar level of difficulty as modeling MNIST with a FC network (far less than modeling CIFAR-10 with a convnet), and Pong on the same order of modeling CIFAR-10.

horizontal line). A dot is darkened signifying the first $d$ that allows a satisfactory performance. We find that the inverted pendulum task is surprisingly easy, with $d_{\text{int100}} = d_{\text{int90}} = 4$, meaning that only four parameters are needed to perfectly solve the problem (see Stanley & Miikkulainen (2002) for a similarly small solution found via evolution). The walking humanoid task is more difficult: solutions are found reliably by dimension 700, a similar complexity to that required to model MNIST with an FC network, and far less than modeling CIFAR-10 with a convnet. Finally, to play Pong on Atari (directly from pixels) requires a network trained in a 6k dimensional subspace, making it on the same order of modeling CIFAR-10. For an easy side-by-side comparison we list all intrinsic dimension values found for all problems in Table 1. For more complete ES results see Sec. S6.2, and Sec. S6.1 for DQN results.

## 4 CONCLUSIONS AND FUTURE DIRECTIONS

In this paper, we have defined the intrinsic dimension of objective landscapes and shown a simple method — random subspace training — of approximating it for neural network modeling problems. We use this approach to compare problem difficulty within and across domains. We find in some cases the intrinsic dimension is much lower than the direct parameter dimension, and hence enable network compression, and in other cases the intrinsic dimension is similar to that of the best tuned models, and suggesting those models are better suited to the problem.

Further work could also identify better ways of creating subspaces for reparameterization: here we chose random linear subspaces, but one might carefully construct other linear or non-linear subspaces to be even more likely to contain solutions. Finally, as the field departs from single stack-of-layers image classification models toward larger and more heterogeneous networks (Ren et al., 2015; Kaiser et al., 2017) often composed of many modules and trained by many losses, methods like measuring intrinsic dimension that allow some automatic assessment of model components might provide much-needed greater understanding of individual black-box module properties.

### ACKNOWLEDGMENTS

The authors gratefully acknowledge Zoubin Ghahramani, Peter Dayan, Sam Greydanus, Jeff Clune, and Ken Stanley for insightful discussions, Joel Lehman for initial idea validation, Felipe Such, Edoardo Conti and Xingwen Zhang for helping scale the ES experiments to the cluster, Vashisht Madhavan for insights on training Pong, Shrivastava Anshumali for conversations about random projections, and Ozan Sener for discussion of second order methods. We are also grateful to Paul Mikesell, Leon Rosenshein, Alex Sergeev and the entire OpusStack Team inside Uber for providing our computing platform and for technical support.

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

# SUPPLEMENTARY INFORMATION FOR:
# MEASURING THE INTRINSIC DIMENSION OF OBJECTIVE LANDSCAPES

## S5 ADDITIONAL MNIST RESULTS AND INSIGHTS

### S5.1 SWEEPING DEPTHS AND WIDTHS; MULTIPLE RUNS TO ESTIMATE VARIANCE

In the main paper, we attempted to find $d_{\text{int90}}$ across 20 FC networks with various depths and widths. A grid sweep of number of hidden layers from $\{1,2,3,4,5\}$ and width of each hidden layer from $\{50,100,200,400\}$ is performed, and all 20 plots are shown in Fig. S6. For each $d$ we take 3 runs and plot the mean and variance with blue dots and blue error bars. $d_{\text{int90}}$ is indicated in plots (darkened blue dots) by the dimension at which the median of the 3 runs passes 90% performance threshold. The variance of $d_{\text{int90}}$ is estimated using 50 bootstrap samples. Note that the variance of both accuracy and measured $d_{\text{int90}}$ for a given hyper-parameter setting are generally small, and the mean of performance monotonically increases (very similar to the single-run result) as $d$ increases. This illustrates that the difference between lucky *vs.* unlucky random projections have little impact on the quality of solutions, while the subspace dimensionality has a great impact. We hypothesize that the variance due to different $P$ matrices will be smaller than the variance due to different random initial parameter vectors $\theta_0^{(D)}$ because there are $dD$ i.i.d. samples used to create $P$ (at least in the dense case) but only $D$ samples used to create $\theta_0^{(D)}$, and aspects of the network depending on smaller numbers of random samples will exhibit greater variance. Hence, in some other experiments we rely on single runs to estimate the intrinsic dimension, though slightly more accurate estimates could be obtained via multiple runs.

In similar manner to the above, in Fig. S7 we show the relationship between $d_{\text{int90}}$ and $D$ across 20 networks but using a per-model, directly trained baseline. Most baselines are slightly below 100% accuracy. This is in contrast to Fig. 3, which used a simpler global baseline of 100% across all models. Results are qualitatively similar but with slightly lower intrinsic dimension due to slightly lower thresholds.

### S5.2 ADDITIONAL DETAILS ON SHUFFLED MNIST DATASETS

Two kinds of shuffled MNIST datasets are considered:

- **The shuffled pixel dataset**: the label for each example remains the same as the normal dataset, but a random permutation of pixels is chosen once and then applied to all images in the training and test sets. FC networks solve the shuffled pixel datasets exactly as easily as the base dataset, because there is no privileged ordering of input dimension in FC networks; all orderings are equivalent.
- **The shuffled label dataset**: the images remain the same as the normal dataset, but labels are randomly shuffled for the entire training set. Here, as in (Zhang et al., 2017), we only evaluate training accuracy, as test set accuracy remains forever at chance level (the training set $X$ and $y$ convey no information about test set $p(y|X)$, because the shuffled relationship in test is independent of that of training).

On the full shuffled label MNIST dataset (50k images), we trained an FC network ($L = 5, W = 400$, which had $d_{\text{int90}} = 750$ on standard MNIST), it yields $d_{\text{int90}} = 190$k. We can interpret this as requiring 3.8 floats to memorize each random label (at 90% accuracy). Wondering how this scales with dataset size, we estimated $d_{\text{int90}}$ on shuffled label versions of MNIST at different scales and found curious results, shown in Table S2 and Fig. S8. As the dataset memorized becomes smaller, the number of floats required to memorize each label becomes larger. Put another way, as dataset size

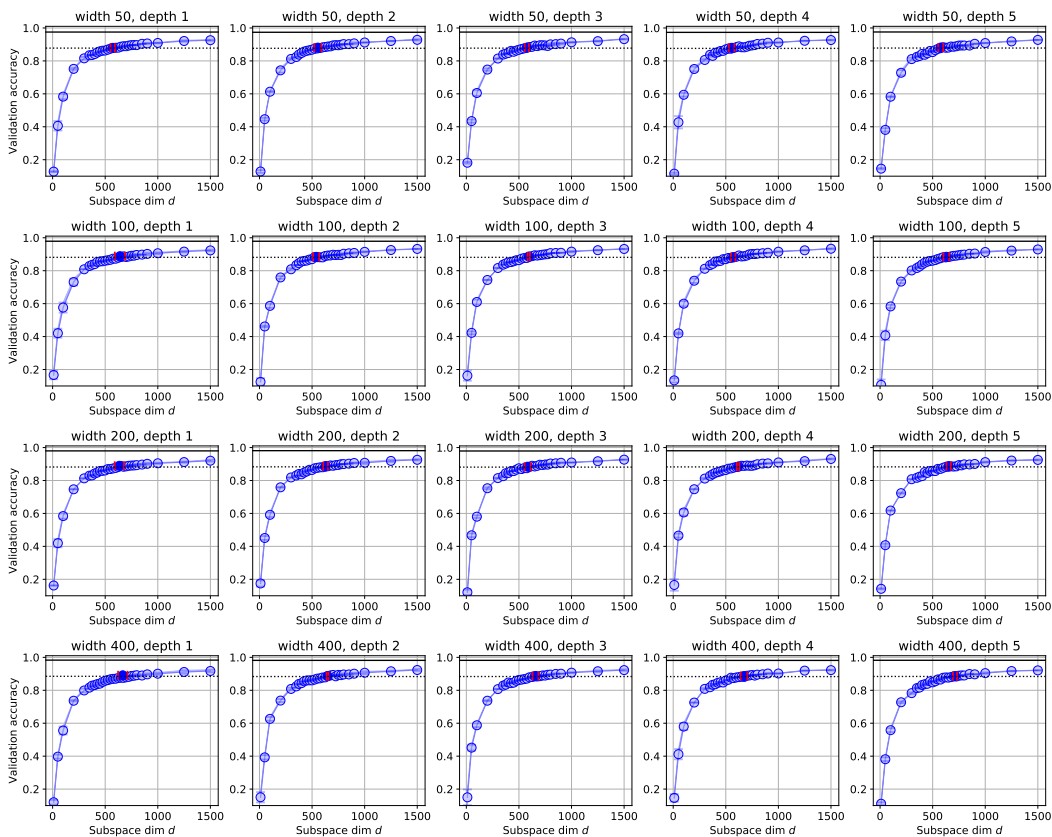

Figure S6: A sweep of FC networks on MNIST. Each column contains networks of the same depth, and each row those of the same number of hidden nodes in each of its layers. Mean and variance at each $d$ is shown by blue dots and blue bars. $d_{int90}$ is found by dark blue dots, and the variance of it is indicated by red bars spanning in the $d$ axis.

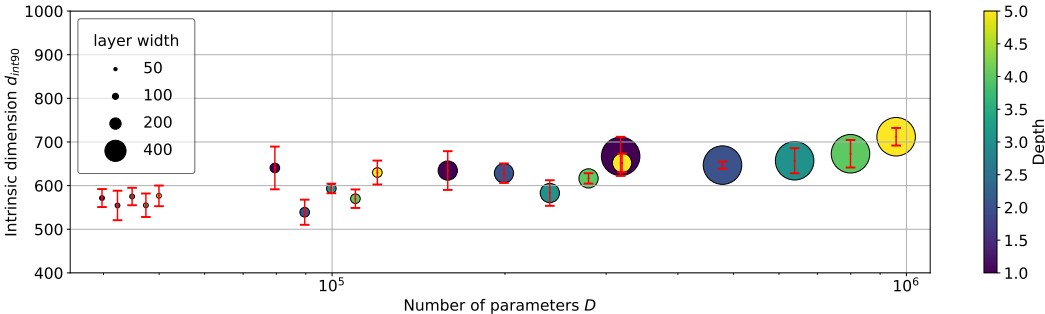

Figure S7: Measured intrinsic dimension $d_{int90}$ vs number of parameters $D$ for 20 models of varying width (from 50 to 400) and depth (number of hidden layers from 1 to 5) trained on MNIST. The vertical red interval is the standard derivation of measured $d_{int90}$. As opposed to Fig. 3, which used a global, shared baseline across all models, here a per-model baseline is used. The number of native parameters varies by a factor of 24.1, but $d_{int90}$ varies by only 1.42. The per-model baseline results in higher measured $d_{int90}$ for larger models because they have a higher baseline performance than the shallower models.

Table S2: $d_{\mathrm{int90}}$ required to memorize shuffled MNIST labels. As dataset size grows, memorization becomes more efficient, suggesting a form of "generalization" from one part of the training set to another, even though labels are random.

| Fraction of MNIST training set | $d_{\mathrm{int90}}$ | Floats per label |
|---|---|---|
| 100% | 190k | 3.8 |
| 50% | 130k | 5.2 |
| 10% | 90k | 18.0 |

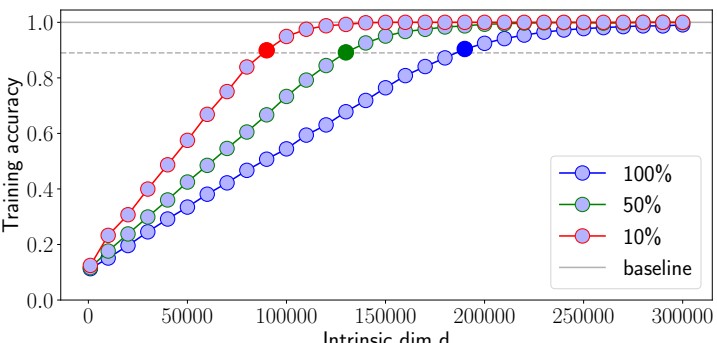

Figure S8: Training accuracy *vs.* subspace dimension $d$ for a FC networks ($W{=}400$, $L{=}5$) trained on a shuffled label version of MNIST containing 100%, 50%, and 10% of the dataset.

increases, the intrinsic dimension also increases, but not as fast as linearly. The best interpretation is not yet clear, but one possible interpretation is that networks required to memorize large training sets make use of shared machinery for memorization. In other words, though performance does not generalize to a validation set, generalization *within* a training set is non-negligible even though labels are random.

### S5.3 TRAINING STABILITY

An interesting tangential observation is that random subspace training can in some cases make optimization more stable. First, it helps in the case of deeper networks. Fig. S9 shows training results for FC networks with up to 10 layers. SGD with step 0.1, and ReLUs with He initialization is used. Multiple networks failed at depths 4, and all failed at depths higher than 4, despite the activation function and initialization designed to make learning stable (He et al., 2015). Second, for MNIST with shuffled labels, we noticed that it is difficult to reach high training accuracy using the direct training method with SGD, though both subspace training with SGD and either type of training with Adam reliably reach 100% memorization as $d$ increases (see Fig. S8).

Because each random basis vector projects across all $D$ direct parameters, the optimization problem may be far better conditioned in the subspace case than in the direct case. A related potential downside is that projecting across $D$ parameters which may have widely varying scale could result in ignoring parameter dimensions with tiny gradients. This situation is similar to that faced by methods like SGD, but ameliorated by RMSProp, Adam, and other methods that rescale per-dimension step sizes to account for individual parameter scales. Though convergence of the subspace approach seems robust, further work may be needed to improve network amenability to subspace training: for example by ensuring direct parameters are similarly scaled by clever initialization or by inserting a pre-scaling layer between the projected subspace and the direct parameters themselves.

### S5.4 THE ROLE OF OPTIMIZERS

Another finding through our experiments with MNIST FC networks has to do with the role of optimizers. The same set of experiments are run with both SGD (learning rate 0.1) and ADAM (learning

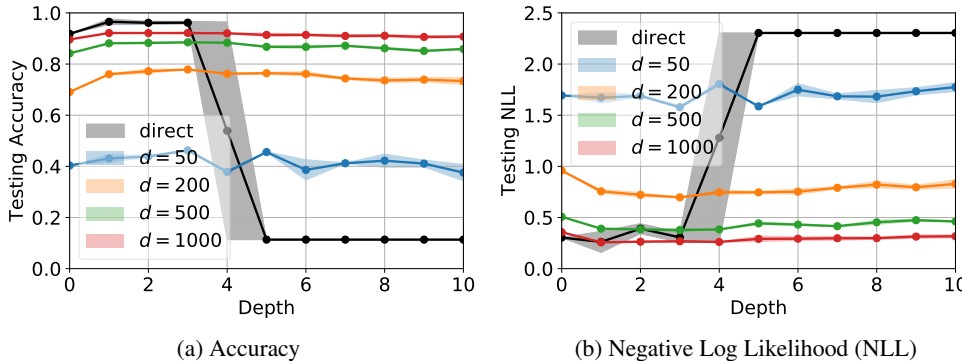

(a) Accuracy            (b) Negative Log Likelihood (NLL)

Figure S9: Results of subspace training versus the number of layers in a fully connected network trained on MNIST. The direct method always fail to converge when $L > 5$, while subspace training yields stable performance across all depths.

rate 0.001), allowing us to investigate the impact of stochastic optimizers on the intrinsic dimension achieved.

The intrinsic dimension $d_{\text{int}90}$ are reported in Fig. S10 (a)(b). In addition to two optimizers we also use two baselines: Global baseline that is set up as 90% of best performance achieved across all models, and individual baseline that is with regards to the performance of the same model in direct training.

## S6 ADDITIONAL REINFORCEMENT LEARNING RESULTS AND DETAILS

### S6.1 DQN EXPERIMENTS

**DQN on Cartpole**  We start with a simple classic control game `CartPole−v0` in OpenAI Gym (Brockman et al., 2016). A pendulum starts upright, and the goal is to prevent it from falling over. The system is controlled by applying a force of LEFT or RIGHT to the cart. The full game ends when one of two failure conditions is satisfied: the cart moves more than 2.4 units from the center (where it started), or the pole is more than 15 degrees from vertical (where it started). A reward of +1 is provided for every time step as long as the game is going. We further created two easier environments `Pole` and `Cart`, each confined by one of the failure modes only.

A DQN is used, where the value network is parameterized by an FC ($L = 2, W = 400$). For each subspace $d$ at least 5 runs are conducted, the mean of which is used to computed $d_{\text{int}90}$, and the baseline is set as 195.0[7]. The results are shown in Fig. S11. The solid line connects mean rewards within a run over the last 100 episodes, across different $d$s. Due to the noise-sensitiveness of RL games the course is not monotonic any more. The intrinsic dimension for `CartPole`, `Pole` and `Cart` is $d_{\text{int}90} = 25, 23$ and $7$, respectively. This reveals that the difficulty of optimization landscape of these games is remarkably low, as well as interesting insights such as driving a cart is much easier than keeping a pole straight, the latter being the major cause of difficulty when trying to do both.

### S6.2 EVOLUTIONARY STRATEGIES (ES) COMPLETE RESULTS

We carry out with ES 3 RL tasks: `InvertedPendulum−v1`, `Humanoid−v1`, `Pong−v0`. The hyperparameter settings for training are in Table S3.

**Inverted pendulum**  The `InvertedPendulum−v1` environment uses the MuJoCo physics simulator (Todorov et al., 2012) to instantiate the same problem as `CartPole−v0` in a realistic setting. We expect that even with richer environment dynamics, as well as a different RL algorithm – ES – the intrinsic dimensionality should be similar. As seen in Fig. 5, the measured intrinsic dimensionality $d_{\text{int}90} = 4$ is of the same order of magnitude, but smaller. Interestingly, although the environment

---

[7]`CartPole−v0` is considered as "solved" when the average reward over the last 100 episodes is 195

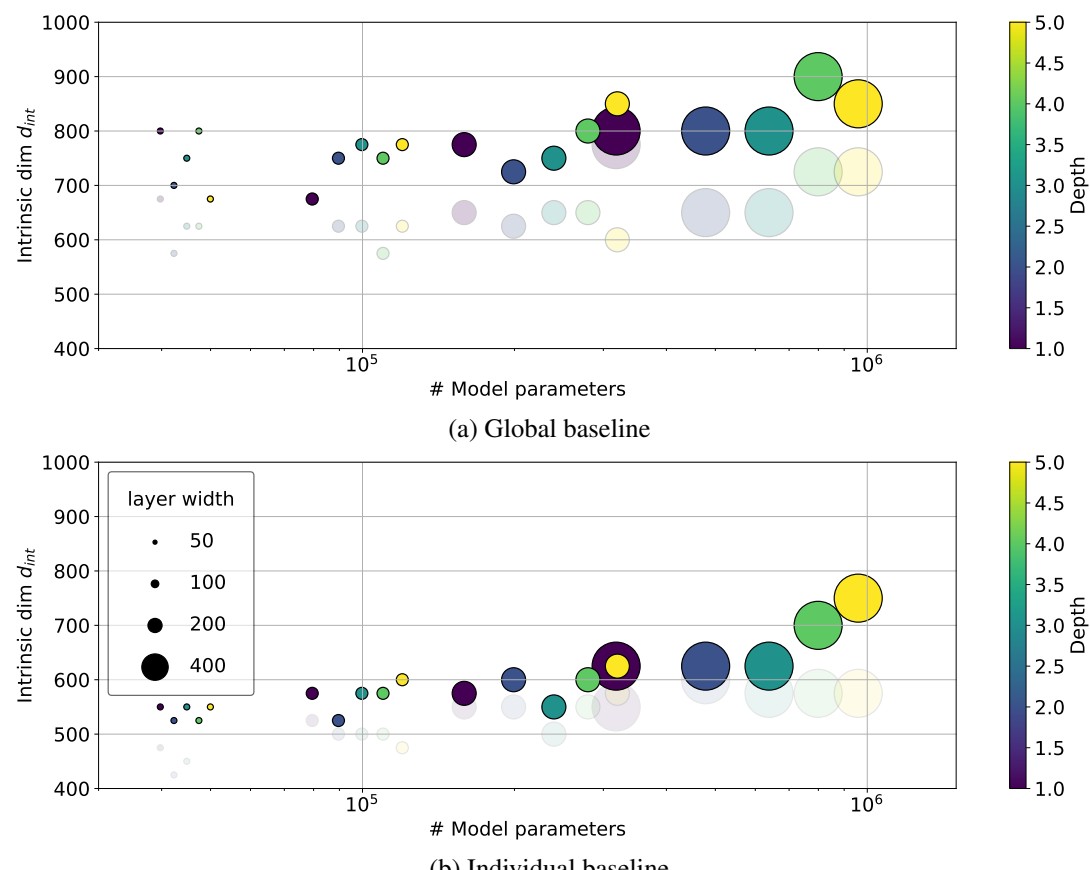

(a) Global baseline

(b) Individual baseline

Figure S10: The role of optimizers on MNIST FC networks. The transparent dots indicate SGD results, and opaque dots indicate Adam results. Adam generally yields higher intrinsic dimensions because higher baselines are achieved, especially when individual baselines in (b) are used. Note that the Adam points are slightly different between Fig. 3 and Fig. S7, because in the former we average over three runs, and in the latter we show one run each for all optimization methods.

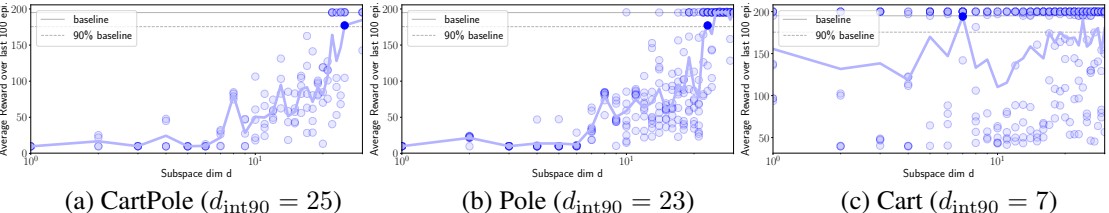

(a) CartPole ($d_{int90} = 25$)      (b) Pole ($d_{int90} = 23$)      (c) Cart ($d_{int90} = 7$)

Figure S11: Subspace training of DQN on CartPole game. Shown as dots are rewards collected through a game run averaged over the last 100 episodes, under each subspace training of DQN, and each game environment. The line connects mean rewards across different $d$s.

dynamics are more complex than in `CartPole−v0`, using ES rather than DQN seems to induce a simpler objective landscape.

**Learning to walk**    A more challenging problem is `Humanoid−v1` in MuJoCo simulator. Intuitively, one might believe that learning to walk is a more complex task than classifying images. Our results show the contrary – that the learned intrinsic dimensionality of $d_{int90} = 700$ is similar to that of MNIST on a fully-connected network ($d_{int90} = 650$) but significantly less than that of even a convnet trained on CIFAR-10 ($d_{int90} = 2,500$). Fig. 5 shows the full results. Interestingly, we

| | $\ell_2$ penalty | Adam LR | ES $\sigma$ | Iterations |
|---|---|---|---|---|
| `InvertedPendulum−v1` | $1 \times 10^{-8}$ | $3 \times 10^{-1}$ | $2 \times 10^{-2}$ | 1000 |
| `Humanoid−v1` | $5 \times 10^{-3}$ | $3 \times 10^{-2}$ | $2 \times 10^{-2}$ | 2000 |
| `Pong−v0` | $5 \times 10^{-3}$ | $3 \times 10^{-2}$ | $2 \times 10^{-2}$ | 500 |

Table S3: Hyperparameters used in training RL tasks using ES. $\sigma$ refers to the parameter perturbation noise used in ES. Default Adam parameters of $\beta_1 = 0.9$, $\beta_2 = 0.999$, $\epsilon = 1 \times 10^{-7}$ were used.

begin to see training runs reach the threshold as early as $d = 400$, with the median performance steadily increasing with $d$.

**Atari Pong**  Finally, using a base convnet of approximately $D = 1M$ in the `Pong−v0` pixels-to-actions environment (using 4-frame stacking). The agent receives an image frame (size of $210 \times 160 \times 3$) and the action is to move the paddle UP or DOWN. We were able to determine $d_{\text{int90}} = 6,000$.

## S7   THREE METHODS OF RANDOM PROJECTION

Scaling the random subspace training procedure to large problems requires an efficient way to map from $\mathbb{R}^d$ into a random $d$-dimensional subspace of $\mathbb{R}^D$ that does not necessarily include the origin. Algebraically, we need to left-multiply a vector of parameters $v \in \mathbb{R}^d$ by a random matrix $M \in \mathbb{R}^{D \times d}$, whose columns are orthonormal, then add an offset vector $\theta_0 \in \mathbb{R}^D$. If the low-dimensional parameter vector in $\mathbb{R}^d$ is initialized to zero, then specifying an offset vector is equivalent to choosing an initialization point in the original model parameter space $\mathbb{R}^D$.

A naïve approach to generating the random matrix $M$ is to use a dense $D \times d$ matrix of independent standard normal entries, then scale each column to be of length 1. The columns will be approximately orthogonal if $D$ is large because of the independence of the entries. Although this approach is sufficient for low-rank training of models with few parameters, we quickly run into scaling limits because both matrix-vector multiply time and storage of the matrix scale according to $\mathcal{O}(Dd)$. We were able to successfully determine the intrinsic dimensionality of MNIST ($d$=225) using a LeNet ($D$=44,426), but were unable to increase $d$ beyond 1,000 when applying a LeNet ($D$=62,006) to CIFAR-10, which did not meet the performance criterion to be considered the problems intrinsic dimensionality.

Random matrices need not be dense for their columns to be approximately orthonormal. In fact, a method exists for "very sparse" random projections (Li et al., 2006), which achieves a density of $\frac{1}{\sqrt{D}}$. To construct the $D \times d$ matrix, each entry is chosen to be nonzero with probability $\frac{1}{\sqrt{D}}$. If chosen, then with equal probability, the entry is either positive or negative with the same magnitude in either case. The density of $\frac{1}{\sqrt{D}}$ implies $\sqrt{D}d$ nonzero entries, or $\mathcal{O}(\sqrt{D}d)$ time and space complexity. Implementing this procedure allowed us to find the intrinsic dimension of $d$=2,500 for CIFAR-10 using a LeNet mentioned above. Unfortunately, when using Tensorflow's `SparseTensor` implementation we did not achieve the theoretical $\sqrt{D}$-factor improvement in time complexity (closer to a constant 10x). Nonzero elements also have a significant memory footprint of 24 bytes, so we could not scale to larger problems with millions of model parameters and large intrinsic dimensionalities.

We need not explicitly form and store the transformation matrix. The Fastfood transform (Le et al., 2013) was initially developed as an efficient way to compute a nonlinear, high-dimensional feature map $\phi(x)$ for a vector $x$. A portion of the procedure involves implicitly generating a $D \times d$ matrix with approximately uncorrelated standard normal entries, using only $\mathcal{O}(D)$ space, which can be multiplied by $v$ in $\mathcal{O}(D \log d)$ time using a specialized method. The method relies on the fact that Hadamard matrices multiplied by Gaussian vectors behave like dense Gaussian matrices. In detail, to implicitly multiply $v$ by a random *square* Gaussian matrix $M$ with side-lengths equal to a power of two, the matrix is factorized into multiple simple matrices: $M = HG\Pi HB$, where $B$ is a random diagonal matrix with entries +-1 with equal probability, $H$ is a Hadamard matrix, $\Pi$ is a random permutation matrix, and $G$ is a random diagonal matrix with independent standard normal entries. Multiplication by a Hadamard matrix can be done via the Fast Walsh-Hadamard Transform

in $\mathcal{O}(d \log d)$ time and takes no additional space. The other matrices have linear time and space complexities. When $D > d$, multiple independent samples of $M$ can be stacked to increase the output dimensionality. When $d$ is not a power of two, we can zero-pad $v$ appropriately. Stacking $\frac{D}{d}$ samples of $M$ results in an overall time complexity of $\mathcal{O}(\frac{D}{d} d \log d) = \mathcal{O}(D \log d)$, and a space complexity of $\mathcal{O}(\frac{D}{d} d) = \mathcal{O}(D)$. In practice, the reduction in space footprint allowed us to scale to much larger problems, including the Pong RL task using a 1M parameter convolutional network for the policy function.

Table S4 summarizes the performance of each of the three methods theoretically and empirically.

|  | Time complexity | Space complexity | $D = 100k$ | $D = 1M$ | $D = 60M$ |
|---|---|---|---|---|---|
| Dense | $\mathcal{O}(Dd)$ | $\mathcal{O}(Dd)$ | 0.0169 s | 1.0742 s* | 4399.1 s* |
| Sparse | $\mathcal{O}(\sqrt{D}d)$ | $\mathcal{O}(\sqrt{D}d)$ | 0.0002 s | 0.0019 s | 0.5307 s* |
| Fastfood | $\mathcal{O}(D \log d)$ | $\mathcal{O}(D)$ | 0.0181 s | 0.0195 s | 0.7949 s |

Table S4: Comparison of theoretical complexity and average duration of a forward+backward pass through $M$ (in seconds). $d$ was fixed to 1% of $D$ in each measurement. $D = 100k$ is approximately the size of an MNIST fully-connected network, and $D = 60M$ is approximately the size of AlexNet. The Fastfood timings are based on a Tensorflow implementation of the Fast Walsh-Hadamard Transform, and could be drastically reduced with an efficient CUDA implementation. Asterisks mean that we encountered an out-of-memory error, and the values are extrapolated from the largest successful run (a few powers of two smaller). For example, we expect sparse to outperform Fastfood if it didn't run into memory issues.

Figure S4 compares the computational time for direct and subspace training (various projections) methods for each update. Our subspace training is more computational expensive, because the subspace training method has to propagate the signals through two modules: the layers of neural networks, and the projection between two spaces. The direct training only propagates signals in the layers of neural networks. We have made efforts to reduce the extra computational cost. For example, the sparse projection less than doubles the time cost for a large range of subspace dimensions.

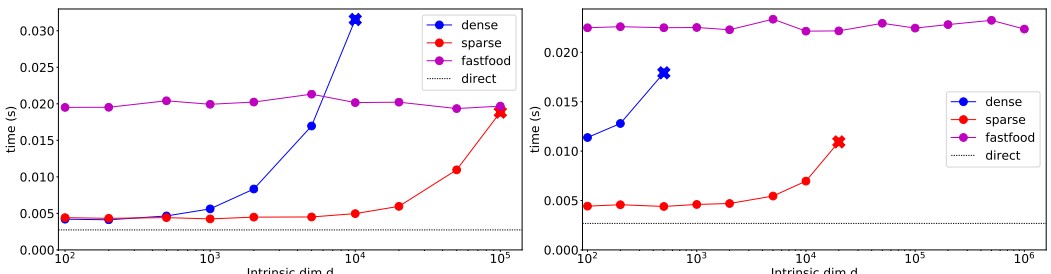

Figure S12: MNIST compute time for direct *vs.* various projection methods for 100k parameters (left) and 1M parameters (right).

## S8  ADDITIONAL CIFAR-10 RESULTS

**FC networks**   We consider the CIFAR-10 dataset and test the same set of FC and LeNet architectures as on MNIST. For FC networks, $d_{\text{int90}}$ values for all 20 networks are shown in Fig. S13 (a) plotted against the native dimension $D$ of each network; $D$ changes by a factor of 12.16 between the smallest and largest networks, but $d_{\text{int90}}$ changes over this range by a factor of 5.0. However, much of this change is due to change of baseline performance. In Fig. S13 (b), we instead compute the intrinsic dimension with respect to a global baseline: 50% validation accuracy. $d_{\text{int90}}$ changes over this range by a factor of 1.52. This indicates that various FC networks share similar intrinsic dimension ($d_{\text{int90}} = 5000 \sim 8000$) to achieve the same level of task performance. For LeNet ($D = 62,006$), the validation accuracy *vs.* subspace dimension $d$ is shown in Fig. S14 (b), the corresponding $d_{\text{int90}} = 2900$. It yields a compression rate of 5%, which is 10 times larger than LeNet on MNIST. It shows that CIFAR-10 images are significantly more difficult to be correctly classified

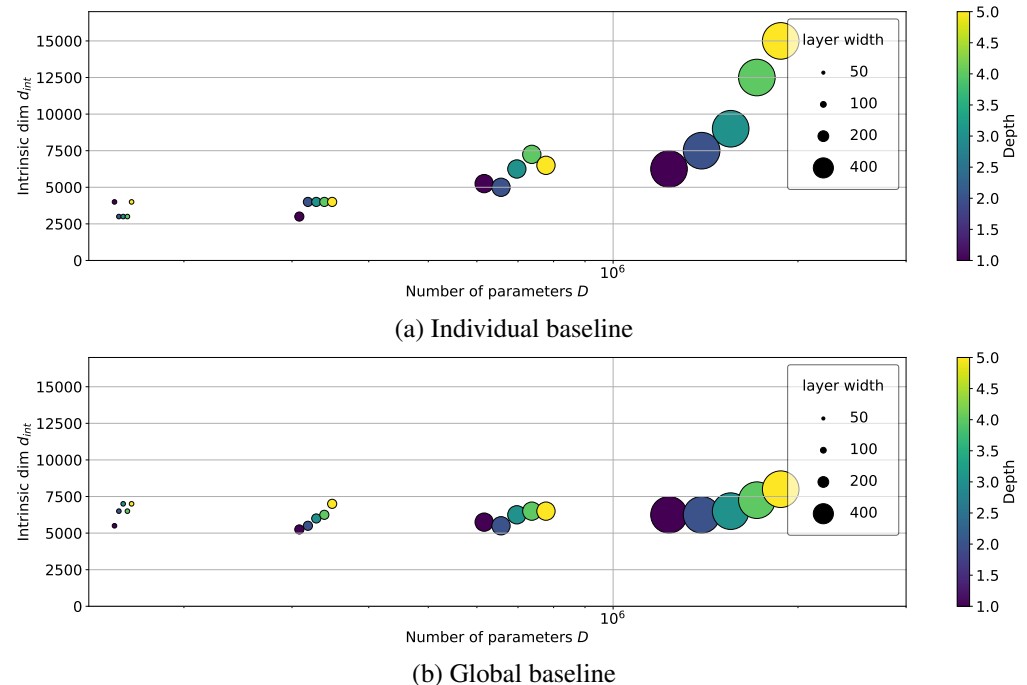

(a) Individual baseline

(b) Global baseline

Figure S13: Intrinsic dimension of FC networks with various width and depth on the CIFAR-10 dataset. In (b), we use a simple 50% baseline globally.

than MNIST. In another word, CIFAR-10 is a harder problem than MNIST, especially given the fact that the notion of "problem solved" (baseline performance) is defined as 99% accuracy on MNIST and 58% accuracy on CIFAR-10. On the CIFAR-10 dataset, as $d$ increases, subspace training tends to overfitting; we study the role of subspace training as a regularizer below.

**ResNet *vs.* LeNet**   We test ResNets, compare to LeNet, and find they make efficient use of parameters. We adopt the smallest 20-layer structure of ResNet with 280k parameters, and find out in Fig. S14 (b) that it reaches LeNet baseline with $d_{\mathrm{int}90} = 1000 \sim 2000$ (lower than the $d_{\mathrm{int}90}$ of LeNet), while takes a larger $d_{\mathrm{int}90}$ ($20,000 \sim 50,000$) to reach reach its own, much higher baseline.

**The role of regularizers**   Our subspace training can be considered as a regularization scheme, as it restricts the solution set. We study and compare its effects with two traditional regularizers with an FC network ($L$=2, $W$=200) on CIFAR-10 dataset, including $\ell_2$ penalty on the weights (*i.e.,* weight decay) and Dropout.

- $\ell_2$ **penalty**   Various amount of $\ell_2$ penalty from $\{10^{-2}, 10^{-3}, 5 \times 10^{-4}, 10^{-4}, 10^{-5}, 0\}$ are considered. The accuracy and negative log-likelihood (NLL) are reported in Fig. S15 (a) (b), respectively. As expected, larger amount of weight decay reduces the gap between training and testing performance for both direct and subspace training methods, and eventually closes the gap (*i.e.,* $\ell_2$ penalty = 0.01). Subspace training itself exhibits strong regularization ability, especially when $d$ is small, at which the performance gap between training and testing is smaller.

- **Dropout**   Various dropout rates from $\{0.5, 0.4, 0.3, 0.2, 0.1, 0\}$ are considered. The accuracy and NLL are reported in Fig. S16. Larger dropout rates reduce the gap between training and testing performance for both direct and subspace training methods. When observing testing NLL, subspace training tends to overfit the training dataset less.

- **Subspace training as implicit regularization**   Subspace training method performs implicit regularization, as it restricts the solution set. We visualized the testing NLL in Fig. S17. Subspace training method outperforms direct method when $d$ is properly chosen (when $\ell_2$ penalty$< 5 \times 10^{-4}$, or dropout rate $< 0.1$), suggesting the potential of this method as a better alternative to traditional regularizers. When $d$ is large, the method also overfits the training

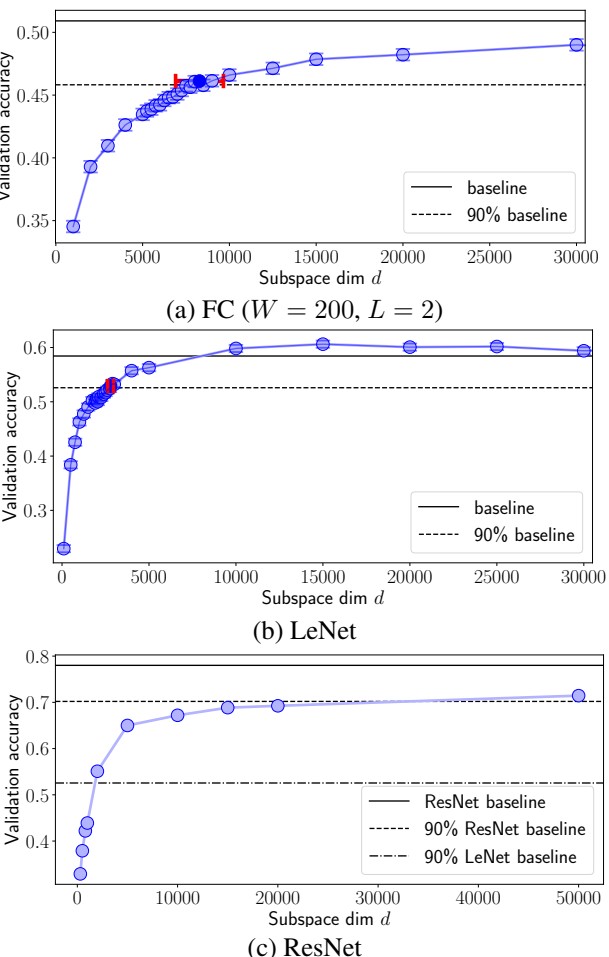

(a) FC ($W = 200$, $L = 2$)

(b) LeNet

(c) ResNet

Figure S14: Validation accuracy of an FC network, LeNet and ResNet on CIFAR with different subspace dimension $d$. In (a)(b), the variance of validation accuracy and measured $d_{int90}$ are visualized as the blue vertical and red horizontal error bars, respectively. Subspace method surpasses the 90% baseline on LeNet at $d$ between 1000 and 2000, 90% of ResNet baseline between 20k and 50k.

dataset. Note that the these methods perform regularization in different ways: weight decay enforces the learned weights concentrating around zeros, while subspace training directly reduces the number of dimensions of the solution space.

## S9  IMAGENET

To investigate even larger problems, we attempted to measure $d_{int90}$ for an ImageNet classification network. We use a relatively smaller network, SqueezeNet by Iandola et al. (2016), with 1.24M parameters. Larger networks suffered from memory issues. A direct training produces Top-1 accuracy of 55.5%. We vary intrinsic dimension from 50k, 100k, 200k, 500k, 800k, and record the validation accuracies as shown in Fig. S18. The training of each intrinsic dimension takes about 6 to 7 days, distributed across 4 GPUs. Due to limited time, training on ImageNet has not yet produced a reliable estimate for $d_{int90}$ except that it is over 500k.

## S10  INVESTIGATION OF CONVOLUTIONAL NETWORKS

Since the learned $d_{int90}$ can be used as a robust measure to study the fitness of neural network architectures for specific tasks, we further apply it to understand the contribution of each component in convolutional networks for image classification task. The convolutional network

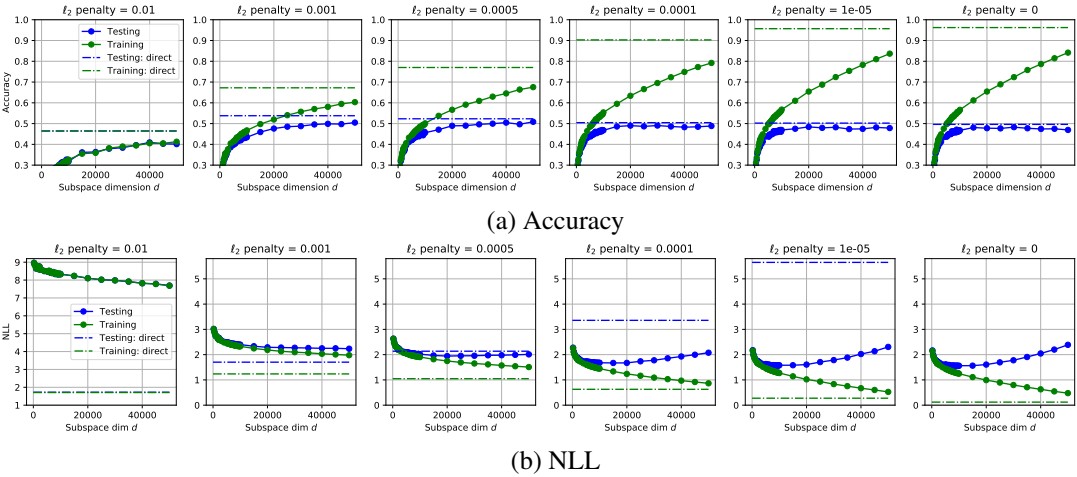

Figure S15: Comparing regularization induced by $\ell_2$ penalty and subspace training. Weight decay interacts with $d_{int90}$ since it changes the objective landscapes through various loss functions.

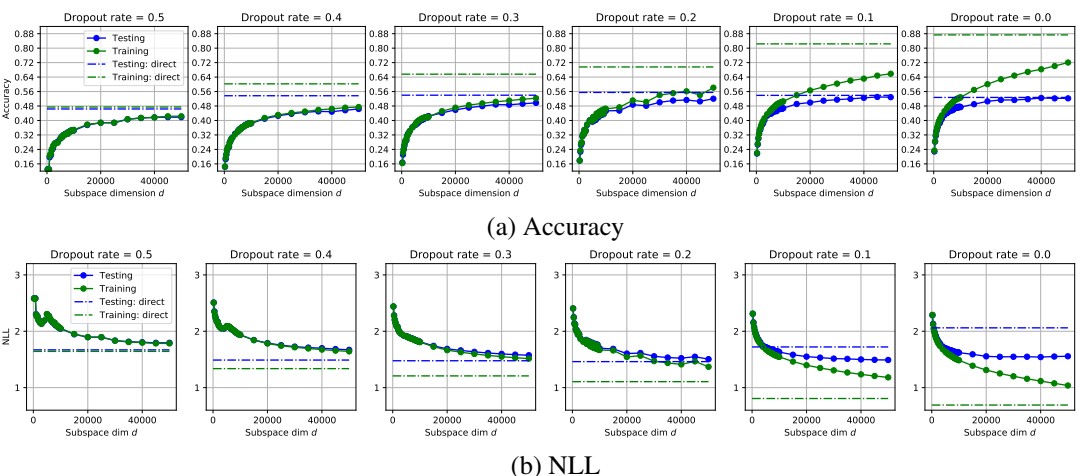

Figure S16: Comparing regularization induced by Dropout and subspace training. Dropout interacts with $d_{int90}$ since it changes the objective landscapes through randomly removing hidden units of the extrinsic neural networks.

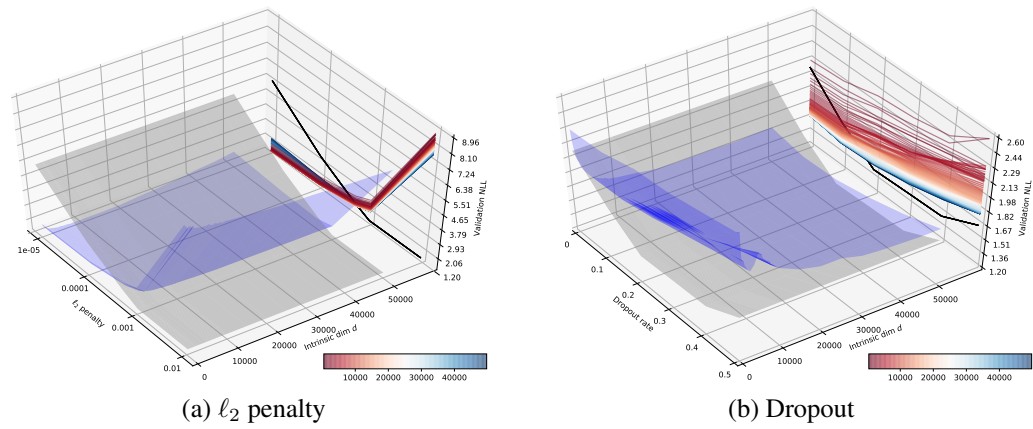

Figure S17: Comparing regularization induced by $\ell_2$ penalty, Dropout and subspace training. The gray surface and black line indicate direct training.

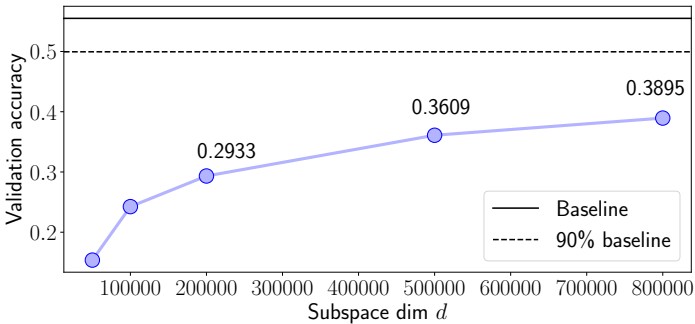

Figure S18: Validation accuracy of SqueezeNet on ImageNet with different $d$. At $d = 500k$ the accuracy reaches 34.34%, which is not yet past the threshold required to estimate $d_{\text{int90}}$.

is a special case of FC network in two aspects: local receptive fields and weight-tying. Local receptive fields force each filter to "look" only at a small, localized region of the image or layer below. Weight-tying enforces that each filter shares the same weights, which reduces the number of learnable parameters. We performed control experiments to investigate the degree to which each component contributes. Four variants of LeNet are considered:

- **Standard LeNet**  6 kernels $(5 \times 5)$ – max-pooling $(2 \times 2)$ – 16 kernels $(5 \times 5)$ – max-pooling $(2 \times 2)$ – 120 FC – 84 FC – 10 FC

- **Untied-LeNet**  The same architecture with the standard LeNet is employed, except that weights are unshared, *i.e.,* a different set of filters is applied at each different patch of the input. For example in Keras, the `LocallyConnected2D` layer is used to replace the `Conv2D` layer.

- **FCTied-LeNet**  The same set of filters is applied at each different patch of the input. we break local connections by applying filters to global patches of the input. Assume the image size is $H \times H$, the architecture is 6 kernels $((2H-1) \times (2H-1))$ – max-pooling $(2 \times 2)$ – 16 kernels $((H-1) \times (H-1))$ – max-pooling $(2 \times 2)$ – 120 FC – 84 FC – 10 FC. The padding type is `same`.

- **FC-LeNet**  Neither local connections or tied weights is employed, we mimic LeNet with its FC implementation. The same number of hidden units as the standard LeNet are used at each layer.

The results are shown in Fig. S19 (a)(b). We set a crossing-line accuracy (*i.e.,* threshold) for each task, and investigate $d_{\text{int90}}$ needed to achieve it. For MNIST and CIFAR-10, the threshold is 90% and 45%, respectively. For the above LeNet variants, $d_{\text{int90}} = 290, 600, 425, 2000$ on MNIST, and $d_{\text{int90}} = 1000, 2750, 2500, 35000$ on CIFAR-10. Experiments show both tied-weights and local connections are important to the model. That tied-weights should matter seems sensible. However, models with maximal convolutions (convolutions covering the whole image) may have had the same intrinsic dimension as smaller convolutions, but this turns out not to be the case.

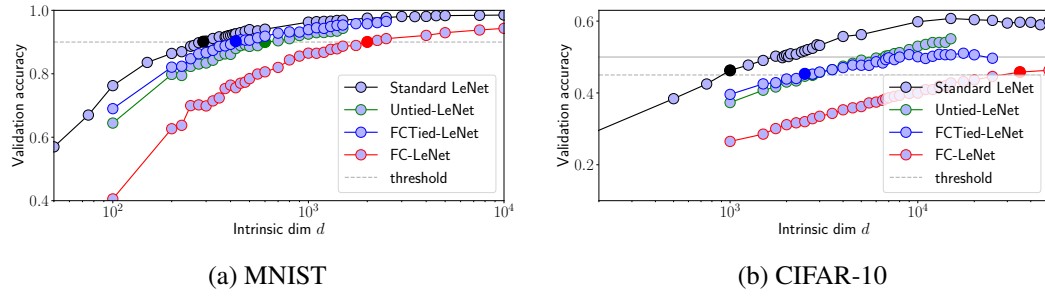

(a) MNIST                                    (b) CIFAR-10

Figure S19: Validation accuracy of LeNet variants with different subspace dimension $d$. The conclusion is that convnets are more efficient than FC nets both due to local connectivity and due to weight tying.

## S11 SUMMARIZATION OF $d_{\text{int}90}$

We summarize $d_{\text{int}90}$ of the objective landscape on all different problems and neural network architectures in Table S5 and Fig. S20. "SP" indicates shuffled pixel, and "SL" for shuffled label, and "FC-5" for a 5-layer FC. $d_{\text{int}90}$ indicates the minimum number of dimensions of trainable parameters required to properly solve the problem, and thus reflects the difficulty level of problems.

Table S5: Intrinsic dimension of different objective landscapes, determined by dataset and network.

| Dataset | Network | $D$ | $d_{\text{int}90}$ |
|---|---|---|---|
| MNIST | FC | 199210 | 750 |
| MNIST | LeNet | 44426 | 275 |
| CIFAR-10 | FC | 1055610 | 9000 |
| CIFAR-10 | LeNet | 62006 | 2900 |
| MNIST-SP | FC | 199210 | 750 |
| MNIST-SP | LeNet | 44426 | 650 |
| MNIST-SL-100% | FC-5 | 959610 | 190000 |
| MNIST-SL-50% | FC-5 | 959610 | 130000 |
| MNIST-SL-10% | FC-5 | 959610 | 90000 |
| ImageNet | SqueezeNet | 1248424 | >500000 |
| CartPole | FC | 199210 | 25 |
| Pole | FC | 199210 | 23 |
| Cart | FC | 199210 | 7 |
| Inverted Pendulum | FC | 562 | 4 |
| Humanoid | FC | 166673 | 700 |
| Atari Pong | ConvNet | 1005974 | 6000 |

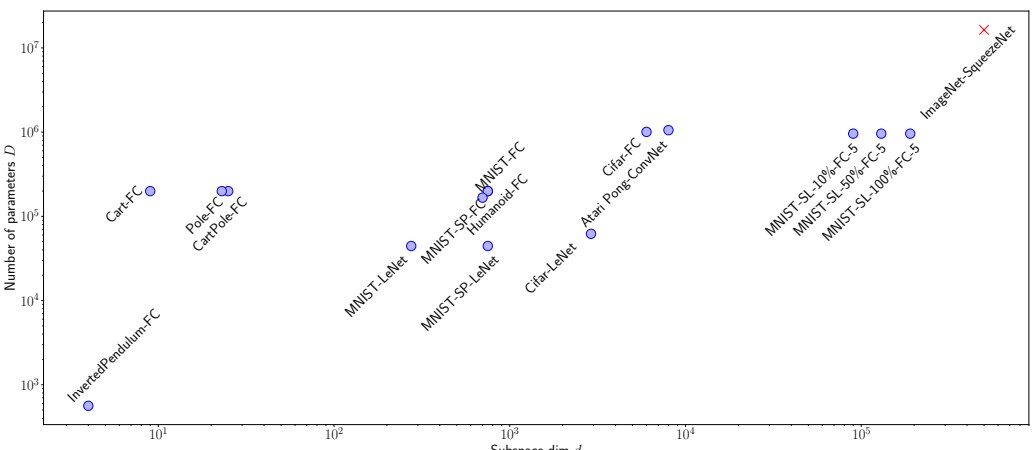

Figure S20: Intrinsic dimension of the objective landscapes created by all combinations of *dataset* and *network* we tried in this paper.

