# OpenReview forum: "Measuring the Intrinsic Dimension of Objective Landscapes"
_ICLR.cc/2018/Conference — Accept (Poster)_

### Official Review · AnonReviewer1 · 2017-11-26
**Good paper**

**Rating:** 7
**Confidence:** 4

**Review:**

[ =============================== REVISION =========================================================]
My questions are answered, paper undergone some revision to clarify the presentation. I still maintain that it is a good paper and argue for acceptance - it provides a witty way of checking whether the network is overparameterized. Mnist  with shuffled labels is a great example that demonstrates the value of the approach, I would though have moved the results of it into the main paper, instead of supplemental materials
[ ======================== END OF REVISION =========================================================]

Authors introduce ransom subspace training (random subspace neural nets) where for a fixed architecture, only a subset of the parameters is trained, and the update for all the parameters is derived via random projection which is fixed for the duration of the training. Using this type of a network, authors introduce a notion of intrinsic dimension of optimization problems - it is minimal dimension of a subset, for which random subset neural net already reaches best (or comparable) performance.
Authors mention that this can be used for compressing networks - one would need to store the seed for the random matrix and the # of params equal to the intrinsic dimension of the net.
They then demonstrate that the intrinsic dimension for the same problem stays the same when different architectures are chosen. Finally they mention neural nets with comparable number of params to intrinsic dimension but that don’t use random subspace trick don’t achieve comparable performance. This does not always hold for CNNs
Model with smaller intrinsic dimension is suggested to be better . They also suggest that intrinsic dimension might be a good approximation to Minimum Description Length metric

My main concern is computational efficiency. They state that if used for compressing, their method  is different from post-train compression, and the authors state that they train once end-to-end. It is indeed the case that once they found model that performs well, it is easy to compress, however they do train a number of models (up to a number of intrinsic dimension) until they get to this admissible model, which i envision would be computationally very expensive.

Questions:
- Are covets always better on MNIST: didn’t understand when authors said that intrinsic dimension of FC on shuffled data stayed the same (why) and then say that it becomes 190K - which one is correct?
- MNIST - state the input dimension size, not clear how you got to that number of parameters overall

---

> ### Author Response · Authors · 2018-01-05
> **Thanks for your kind and helpful comments**
>
> Thanks for your kind and helpful comments! Replies below:
>
> > My main concern is computational efficiency [to compute intrinsic dimension]
>
> You are correct that the computational cost can be high to obtain the intrinsic dimension. In practice, one may find the intrinsic dimension by either (a) running many training runs at different dimensions in parallel (low wall clock time, as we used here), or (b) using binary search (low total amount of computation). In the latter case, the precision of the measured intrinsic dimension is then related to the number of iterations in binary search. Faster methods to obtain intrinsic dimension could be an interesting direction of future work. Slight inefficiency notwithstanding, we see the approach as valuable overall for the insight into network behavior that it provides, even if this insight comes at a cost of a binary search or use of a cluster.
>
> > Are covets always better on MNIST:
>
> Yes, convnets are always better on MNIST than FC networks, EXCEPT in the case where the pixel order is shuffled (see more on this below).
>
> > didn’t understand when authors said that intrinsic dimension of FC on shuffled data stayed the same (why) and then say that it becomes 190K - which one is correct?
>
> Sorry for this confusion, and thanks for pointing it out. The setting will make sense for those that know the Zhang et al 2017 ICLR paper in detail, but for those not familiar, the wording as submitted was indeed quite confusing. We’ve updated the text to explain the situation more completely (See Section S5.3).
>
> To wit, there are two different shuffled MNIST datasets:
>
> (a) a “shuffled pixel” dataset in which the label for each example remains the same as the normal dataset, but a random permutation of pixels is chosen once and then applied to all images in the training and test sets. FC networks solve the shuffled pixel datasets exactly as easily as the base dataset, because there is no privileged ordering of input dimension in FC networks -- all orderings are equivalent. Convnets suffer here because they expect local structure but the local structure was destroyed by shuffling pixel locations.
>
> (b) a “shuffled label” dataset in which the images remain the same as the base dataset, but training labels are randomly shuffled for the entire training set. Here, as in [Zhang et al, ICLR 2017], we only evaluate training accuracy, as test set accuracy remains forever at chance level (the training set X and y convey no information about test set p(y|X), because test set labels are shuffled independent of the training set).
>
> The intrinsic dimension of FC nets on the shuffled-label training set becomes huge: 190K. This is because the network must memorize every training set label (well, 90% of them), and the capacity required to do so is large. This illustrates cleanly an important concept: while the standard dataset and the shuffled-label dataset are of exactly the same size, containing exactly the same bits, and providing exactly the same number of constraints on the learned p(y|X), the random version contains much higher entropy, a fact which we can *measure* by computing intrinsic dimension of neural network! In the standard dataset, the constraints imposed by the image-label pairs in the same class are very similar, thus parameters representing those constraints can be shared. The number of unique constraints is small. In contrast, the shuffled-label dataset fully randomizes the well-structured relationship among image-label pairs. Each randomized pair provides a unique constraint for the model, and the neural network has to be optimized to satisfy all the unique constraints. Hence, the number of unique constraints is very large. The number of unique constraints reflects the intrinsic dimension we obtained for each dataset.
>
> > MNIST - state the input dimension size, not clear how you got to that number of parameters overall
>
> Thanks for your careful study; it was a typo! the FC network size is 784-200-200-10 (200, not 400, as we had stated). The total number of parameters (including “+ 1” for biases) is (784 + 1) * 200 + (200 + 1) * 200 + (200 + 1) * 10 = 199210. The draft has been updated to fix this.

---

### Official Review · AnonReviewer3 · 2017-11-26
**An proposal that reduces the degree of freedom in deep learning**

**Rating:** 6
**Confidence:** 2

**Review:**

While deep learning usually involves estimating a large number of variable, this paper suggests to reduce its number by assuming that these variable lie in a low-dimensional subspace. In practice, this subspace is chosen randomly. Simulations show the promise of the proposed method.  In particular, figure 2 shows that the number of parameters could be greatly reduced while keeping 90% of the performance; and figure 4 shows that this method outperforms the standard method. The method is clearly written and the idea looks original.

A con that I have is about the comparison in figure 4. While the proposed subspace method might have the same number of parameters as the direct method, I wonder if it is a fair comparison since the subspace method could still be more computational expensive, due to larger number of latent variables.

---

> ### Author Response · Authors · 2018-01-05
> **Thanks for your kind and helpful comments**
>
> Thanks for your kind and helpful comments! A few replies and thoughts are below:
>
> > ...this paper suggests to reduce its number by assuming that these variable lie in a low-dimensional subspace. In practice, this subspace is chosen randomly.
>
> Not to belabor minutiae, but it may be worth discussing a few subtleties here in case any were not already clear. First, the paper doesn’t assume that parameters lie in a low-dimensional subspace; instead, it asks whether they happen to (it turns out they often do) and whether if they do, we could measure the dimensionality of that subspace using a simple approach -- by intersecting the solution space with random subspaces (this does seem to work). So there are two subspaces under consideration: the subspace of solutions, which is certainly far more structured than random, and the subspace in which we search for intersection, which we do choose to be random. For example, in the Figure 1 toy example, the solution space is 990 dimensional and highly structured, but the 10 dimensional subspace in which we find intersection is random.
>
> > Simulations show the promise of the proposed method.  In particular, figure 2 shows that the number of parameters could be greatly reduced while keeping 90% of the performance; and figure 4 shows that this method outperforms the standard method. The method is clearly written and the idea looks original.
>
> Indeed, the random projection method “outperforms the standard method” if we consider more parsimonious models to outperform those with more parameters. However, note that while this is a fun by-product of the approach, we’d like to emphasize that the primary importance of the work is that it provides a tool that can be used to analyze and measure network behavior. By using random projections, we obtain a window into the complex, high dimensional objective landscape that wasn’t previously reported, and we think this will be quite useful for the field!
>
> We’ve rewritten parts of the introduction to emphasize more clearly that our paper is not primarily one proposing a better model, but a paper providing insights into network properties.
>
> > A con that I have is about the comparison in figure 4. While the proposed subspace method might have the same number of parameters as the direct method, I wonder if it is a fair comparison since the subspace method could still be more computationally expensive, due to larger number of latent variables.
>
> Indeed, the subspace method is always at least a little more computationally expensive, though minimally so. And, as mentioned above, the paper isn’t so much about fast tricks for better training as it about teasing out subtleties of high dimensional landscapes. This latter undertaking would be worth it even if computationally inconvenient; it just so happens that the approach ends up being computationally reasonable compared to the cost of ordinary training!

---

### Official Review · AnonReviewer2 · 2017-11-29
**ICLR 2018 official review (Reviewer 2)**

**Rating:** 7
**Confidence:** 3

**Review:**

This paper proposes an empirical measure of the intrinsic dimensionality of a neural network problem. Taking the full dimensionality to be the total number of parameters of the network model, the authors assess intrinsic dimensionality by randomly projecting the network to a domain with fewer parameters (corresponding to a low-dimensional subspace within the original parameter), and then training the original network while restricting the projections of its parameters to lie within this subspace. Performance on this subspace is then evaluated relative to that over the full parameter space (the baseline). As an empirical standard, the authors focus on the subspace dimension that achieves a performance of 90% of the baseline. The authors then test out their measure of intrinsic dimensionality for fully-connected networks and convolutional networks, for several well-known datasets, and draw some interesting conclusions.

Pros:

* This paper continues the recent research trend towards a better characterization of neural networks and their performance. The authors show a good awareness of the recent literature, and to the best of my knowledge, their empirical characterization of the number of latent parameters is original.

* The characterization of the number of latent variables is an important one, and their measure does perform in a way that one would intuitively expect. For example, as reported by the authors, when training a fully-connected network on the MNIST image dataset, shuffling pixels does not result in a change in their intrinsic dimensionality. For a convolutional network the observed 3-fold rise in intrinsic dimension is explained by the authors as due to the need to accomplish the classification task while respecting the structural constraints of the convnet.

* The proposed measures seem very practical - training on random projections uses far fewer parameters than in the original space (the baseline), and presumably the cost of determining the intrinsic dimensionality would presumably be only a fraction of the cost of this baseline training.

* Except for the occasional typo or grammatical error, the paper is well-written and organized. The issues are clearly identified, for the most part (but see below...).

Cons:

* In the main paper, the authors perform experiments and draw conclusions without taking into account the variability of performance across different random projections. Variance should be taken into account explicitly, in presenting experimental results and in the definition and analysis of the empirical intrinsic dimension itself. How often does a random projection lead to a high-quality solution, and how often does it not?

* The authors are careful to point out that training in restricted subspaces cannot lead to an optimal solution for the full parameter domain unless the subspace intersects the optimal solution region (which in general cannot be guaranteed). In their experiments (FC networks of varying depths and layer widths for the MNIST dataset), between projected and original solutions achieving 90% of baseline performance, they find an order of magnitude gap in the number of parameters needed. This calls into question the validity of random projection as an empirical means of categorizing the intrinsic dimensionality of a neural network.

* The authors then go on to propose that compression of the network be achieved by random projection to a subspace of dimensionality greater than or equal to the intrinsic dimension. However, I don't think that they make a convincing case for this approach. Again, variation is the difficulty: two different projective subspaces of the same dimensionality can lead to solutions that are extremely different in character or quality. How then can we be sure that our compressed network can be reconstituted into a solution of reasonable quality, even when its dimensionality greatly exceeds the intrinsic dimension?

* The authors argue for a relationship between intrinsic dimensionality and the minimum description length (MDL) of their solution, in that the intrinsic dimensionality should serve as an upper bound on the MDL. However they don't formally acknowledge that there is no standard relationship between the number of parameters and the actual number of bits needed to represent the model - it varies from setting to setting, with some parameters potentially requiring many more bits than others. And given this uncertain connection, and given the lack of consideration given to variation in the proposed measure of intrinsic dimensionality, it is hard to accept that "there is some rigor behind" their conclusion that LeNet is better than FC networks for classification on MNIST because its empirical intrinsic dimensionality score is lower.

* The experimental validation of their measure of intrinsic dimension could be made more extensive. In the main paper, they use three image datasets - MNIST, CIFAR-10 and ImageNet. In the supplemental information, they report intrinsic dimensions for reinforcement learning and other training tasks on four other sets.

Overall, I think that this characterization does have the potential to give insights into the performance of neural networks, provided that variation across projections is properly taken into account. For now, more work is needed.

====================================================================================================
Addendum:

The authors have revised their paper to take into account the effect of variation across projections, with results that greatly strengthen their results and provide a much better justification of their approach. I'm satisfied too with their explanations, and how they incorporated them into their revised version. I've adjusted my rating of the paper accordingly.

One point, however: the revisions seem somewhat rushed, due to the many typos and grammatical errors in the updated sections. I would like to encourage the authors to check their manuscript once more, very carefully, before finalizing the paper.
====================================================================================================

---

> ### Author Response · Authors · 2018-01-05
> **Thanks for the valuable, critical feedback**
>
> Thanks for taking the time to provide such thorough comments and such detailed feedback. Much of the feedback alludes to elements of the work that had been critically omitted. We’ve since updated the paper to include 7 missing or modified sections and experiments as described below.
>
> > [Summary and Pros]
>
> Thanks kindly!
>
> > Cons:
>
> (We’ve slightly reordered these in response)
>
> > In the main paper, the authors perform experiments and draw conclusions without taking into account the variability of performance across different random projections. Variance should be taken into account explicitly, in presenting experimental results and in the definition and analysis of the empirical intrinsic dimension itself. How often does a random projection lead to a high-quality solution, and how often does it not?
>
> This is an important question. Initial experiments showed that random projections of the sizes under consideration -- e.g. through a dense random matrix with shape (750 by 200,000) -- contain so many IID random elements (e.g. 150,000,000) that the difference between the luckiest random projection and the least lucky random projection among N = 10 or 20 was tiny and thus possible to ignore. However, we should definitely have described these initial experiments in the paper and otherwise justified the assumption.
>
> We’ve rectified this omission by (a) repeating all FC MNIST experiments three times each and including error bars both on each individual measurement of performance at a given dimension (see, for example, vertical error bars on each dot in the updated Fig S6), and (b) by using these multiple measurements to produce a bootstrapped estimate of the error bars on measurements of intrinsic dimension (see, for example, horizontal error bars in the updated Fig S6).
>
> As can be seen in Fig S6, the variance of performance for a given subspace dimension is very small, and the mean of performance monotonically increases (very similar to one run result). It indicates the luckiness in random projection has a little impact on the quality of solutions, while the subspace dimensionality has a great impact on the quality of solutions.
>
> We further repeat our experiments three times for more network architectures and datasets, and report the mean and standard deviation (std) on bootstrapped samples.
>
> FC (depth=2, width=200) on MNIST:  mean=802.25, std=67.83
> LeNet on MNIST:  mean=290.0, std=0.0
> FC (depth=2, width=200) on CIFAR: mean=8277.5, std=1378.36
> LeNet on CIFAR: mean=2840.0, std=120.0
>
> The std here is of our measurement of the intrinsic dimension based on one-run result. Even considering the interval report here, the numbers can still serve as sufficient evidences for the main interest of paper: providing the insights of understanding network behavior. Hence, we can rely on one run result for fast compute of  the intrinsic dimension, though slightly more accurate solutions can be obtained via multiple runs and refined interval of subspace dimensionality. We’ve explained this justification in Section S5 and Fig S6 in Supplementary Material (see updated draft).
>
> Further, in the cases where the randomness of the learning process has large performance variance (apart from the random projection), we have adjusted and clarified our definition of “intrinsic dimension” to take account of the variance. In reinforcement learning tasks, where the large randomness of the tasks themselves leads to very different performance in different runs, we performed the experiments multiple times, and defined the intrinsic dimension as the dimension at which the mean reward crossed the threshold, where the mean reward is averaged over 30 runs for a given subspace dimension. (Details given in Section 3.3)
>
> > The authors then go on to propose that compression of the network… two different projective subspaces of the same dimensionality can lead to solutions that are extremely different in character or quality.
>
> As now more carefully quantified above, the performance variation across different random projections is minimal. (Also: see next section).
>
> > How then can we be sure that our compressed network can be reconstituted into a solution of reasonable quality, even when its dimensionality greatly exceeds the intrinsic dimension?
>
> We’re not training full networks, compressing them, and then reconstituting them into networks we hope will happen still to perform well (such an approach could indeed fail). Instead, networks are trained directly in the compressed space (“end to end”) and evaluated during training and validation exactly as they would be in production. So training and validation accuracies may be interpreted as faithfully as one would in a normal training scenario.

---

> > ### Author Response · Authors · 2018-01-05
> > **Continue due to the character limit**
> >
> > > The authors are careful to point out...  [T]hey find an order of magnitude gap in the number of parameters needed. This calls into question the validity of random projection as an empirical means of categorizing the intrinsic dimensionality of a neural network.
> >
> > Perhaps we misunderstand part of this objection, but if not, we believe the approach is still quite defensible. We may see this from two directions: empirical and theoretical.
> >
> > Empirical: trying many different random projections tends to produce very similar results (thanks for the suggestion to report this directly!)
> >
> > Theoretical: given two random hyperplanes of dimension m and n in a space of dimension D, m and n will intersect with probability 0 if m + n < D and probability 1 if m + n >= D. The transition from “almost surely will not intersect” to “almost surely will intersect” is sudden. This result is for the intersection of two random hyperplanes, not for the intersection of a structured solution set and a random hyperplane, but we expect (and above measure) similar sudden transitions for actual solution sets.
> >
> > > The authors argue for a relationship between intrinsic dimensionality and the minimum description length (MDL) ...However there is no standard relationship between the number of parameters and the actual number of bits
> >
> > We just give a loose upper bound: one where each parameter -- native or subspace -- requires 32 bits to represent in floating point format. It’s true that some parameters may be represented using far fewer bits, but this does not make the upper bound any less valid. We’ve clarified that the upper bound refers to 32 bit floats, which were used for all experiments.
> >
> > > [More extensive experimental validation using other datasets]
> >
> > The results presented in the paper (not including replicates re-run thanks to the above suggestions) derive from over 10,000 experimental runs from seven datasets/environments. Generally speaking we’d love to include even more, but we believe the theory and method as presented has been sufficiently validated through experiments. We certainly do believe that researchers will be able to generate great insights by applying these methods to further datasets, but at this point we feel it is defensible to delegate that to future work.
> >
> > Nonetheless, we note extra experimentation added to the current draft after the initial submission:
> >
> > In Section S8 (paragraph “The role of regularizers”), we investigate the regularization ability of subspace training vs traditional regularizers (weight decay and Dropout) and their combinations. It is shown that subspace training itself has strong regularization ability by reducing the dimensionality of solution set, and its combination with traditional regularizers can further improve validation performance. Regarding intrinsic dimension, stronger traditional regularizers generally lead to slightly larger measured intrinsic dimension, as regularizers restrict the expressiveness of the model, which must be overcome by extra dimensions.
> > In Section S10, we apply intrinsic dimension to further understand the contribution of each component in convolutional networks for image classification task: local receptive fields and weight-tying (i.e., the two aspects that a convolutional network is a special case of a FC network)
> >
> > All told: again, thanks for the helpful, critical feedback! We think the paper as amended is much stronger than it was on submission and hope you will agree.

---

### Decision · Program_Chairs · 2018-01-29
**ICLR 2018 Conference Acceptance Decision**

**Decision:**

Accept (Poster)

**Comment:**

The authors make an empirical study of the "dimension" of a neural net optimization problem, where the "dimension" is defined by the minimal random linear parameter subspace dimension where a (near) solution to the problem is likely to be found.   I agree with reviewers that in light of the authors' revisions, the results are interesting enough to be presented at the conference.